SPECIAL ISSUE
LIFELONG DEVELOPMENT

# CSF1R ligands promote microglial proliferation but are not the sole regulators of developmental microglial proliferation

Brady P. Hammond[1], Sameera Zia[1], Eugene Hahn[1], Margarita Kapustina[2], Tristan Lange[1], Sarah Friesen[1], Rupali Manek[1], Kelly V. Lee[1], Adrian Castellanos-Molina[3], Floriane Bretheau[3], Mark S. Cembrowski[2,4], Bradley J. Kerr[1,6,7], Steve Lacroix[3,5] and Jason R. Plemel[1,7,8,9,10,*]

## ABSTRACT

Microglia – the predominant immune cells of the brain and spinal cord – perform essential functions for the development and maintenance of the central nervous system, contingent upon the regulated developmental proliferation of microglia. However, the factor(s) that regulate microglial proliferation remain unclear. Here, we confirmed the timeline of developmental proliferation and used bioinformatics to identify potential signalling onto microglia in mouse from datasets collected at an age of high developmental microglial proliferation. Of the predicted factors, we found that colony stimulating factor 1 receptor (CSF1R) ligands boosted proliferation *in vitro* and were increasingly expressed in the brain across development with each displaying a distinct regional and temporal expression pattern. However, we did not observe a coincident alteration to CSF1R ligand levels in a model of abnormal developmental proliferation. Together, although CSF1R ligands can promote microglial proliferation in culture, their developmental expression patterns suggest that they function alongside other unknown factors to regulate developmental microglial proliferation.

KEY WORDS: Microglia, Brain, Development, Proliferation, CSF1R, CSF1, IL34, Mouse

## INTRODUCTION

The appropriate development of the brain and spinal cord lays the foundation for complex neurological function. This development encompasses the expansion of, and interactions between, diverse cell populations. Of these expanding cell populations, microglia –

[1]Neuroscience and Mental Health Institute, University of Alberta, Edmonton, T6G 2R3, Canada. [2]Department of Cellular and Physiological Sciences, Life Sciences Institute, University of British Columbia, Vancouver, V6T 1Z3, Canada. [3]Axe Neurosciences du Centre de Recherche du Centre hospitalier universitaire (CHU) de Québec, Université Laval, Québec City, G1V 0E8, Canada. [4]Djavad Mowafaghian Centre for Brain Health, University of British Columbia, Vancouver, V6T 1Z3, Canada. [5]Département de Médecine Moléculaire de l'Université Laval, Québec City, G1V 0E8, Canada. [6]Department of Anesthesiology and Pain Medicine, University of Alberta, Edmonton, T6G 2R3, Canada. [7]Multiple Sclerosis Centre, University of Alberta, Edmonton, T6G 2R3, Canada. [8]Department of Medicine, Division of Neurology, University of Alberta, Edmonton, T6G 2R3, Canada. [9]Department of Medical Microbiology and Immunology, University of Alberta, Edmonton, T6G 2R3, Canada. [10]Li Ka Shing Institute of Virology, Edmonton, T6G 2R3, Canada.

*Author for correspondence ( jrplemel@ualberta.ca)

 J.R.P., 0000-0003-1385-1464

the predominant immune cell of the central nervous system (CNS) – perform integral functions in development (Li and Barres, 2018; Thion et al., 2018). Microglia are derived from erythromyeloid progenitors of the embryonic yolk sac that seed the CNS in embryogenesis (Ginhoux et al., 2010). After their initial seeding, microglia proliferate until they reach a stable, largely non-proliferative, density in the second postnatal week in mice (Ginhoux et al., 2010; Kim et al., 2015; Nikodemova et al., 2015). After this initial proliferative period, microglia are maintained throughout life by a process of self-renewal (Askew et al., 2017; Tay et al., 2017). In human development, the timeline of microglial proliferation is less clear, although proliferation may account, at least in part, for the increase in microglial density observed from gestational week 4.5 onwards until full-term (Andjelkovic et al., 1998; Esiri et al., 1991; Hutchins et al., 1990; Rezaie et al., 2005). Microglia reach an apparent stable density pre-term and develop a ramified, mature morphology just before full-term is reached (Esiri et al., 1991).

During development, microglia eliminate synaptic material (Paolicelli et al., 2011; Parkhurst et al., 2013; Schafer et al., 2012; Squarzoni et al., 2014) and myelin (Djannatian et al., 2023; Hughes and Appel, 2020). However, whether microglia actively sculpt developing neural circuitry remains unclear and has been reviewed elsewhere (Eyo and Molofsky, 2023; Pereira-Iglesias et al., 2025). A disruption to such microglial dynamics, whether transient or chronic, fundamentally alters developing neural circuitry and activity (Paolicelli et al., 2011; Schafer et al., 2012), although the direct consequences of microglial disruption remain unclear. It may be that the resultant alterations to neural circuitry and activity that stem from microglial disruption have a long-term impact on behaviour. For example, early life stress induced by maternal deprivation in mice throughout the first postnatal weeks yields a transient elevation in microglial densities that normalizes to baseline levels in adulthood. Such stress likewise induces long-term changes in microglial transcriptional state and phagocytic capacity, which, given their role in the uptake of synaptic material and myelin throughout development, may contribute to the long-term depressive-like and anxiety-like behaviours of these mice in adulthood (Delpech et al., 2016). Thus, the appropriate development of microglia facilitates their functions in the early life CNS and potentially beyond.

The temporal restriction of microglial proliferation to development may present a layer of regulation that enables microglia to execute their functions. Outside of the robust proliferative capacity of microglia during development, microglia in the adult CNS are only minimally proliferative at homeostasis, although their proliferative capacity may be reawakened in injurious or diseased states (Hammond et al., 2021). Injury or disease often promotes microglial proliferation and impacts their functions. For example, microglia proliferate after peripheral nerve injury and limiting this

proliferation prevents neuropathic pain (Gu et al., 2016), suggesting that microglial proliferation is linked to the initiation of this pain state. In contrast, boosting microglial proliferation after spinal cord injury facilitates scar formation and restricts neurodegeneration (Bellver-Landete et al., 2019). Several factors have been linked to altered microglial proliferation in a variety of conditions (Hammond et al., 2021), but many of these are correlational or may indirectly influence proliferation. Whether a mitogen – a factor that directly boosts proliferation – directly promotes microglial proliferation in development remains unclear. Here, we aimed to identify the mitogen(s) that drive microglial proliferation during development.

To investigate developmental microglial proliferation, we established a histological timeline of microglial proliferation across development in mice and identified high levels of proliferation in the first postnatal week and negligible levels after the second postnatal week. To identify potential mitogens, we used an unbiased receptor–ligand bioinformatic tool on published single-cell RNA-sequencing (scRNAseq) datasets in development and tested the predicted factors on purified primary microglial cultures to evaluate their mitogenicity. Of the tested factors, only colony-stimulating factor 1 receptor (CSF1R) ligands boosted microglial proliferation. We then tracked changes of these mitogens during brain development and found that expression pattern of these factors did not precisely align with

the timeline of microglial proliferation. We found that microglial proliferation was enhanced in the absence of IL1 cytokines, yet this enhanced proliferation did not correlate with elevated levels of CSF1R ligands. Together, these data show that, although CSF1R ligands are microglial mitogens in culture, other factors that regulate the developmental proliferation of microglia remain to be identified, including those altered in the absence of IL1 cytokines.

## RESULTS

### Microglia proliferate robustly prior to postnatal day 14

We hypothesized that microglial proliferation in development results from a mitogen that directly promotes microglial proliferation. To explore the postnatal proliferation of microglia, we collected and sectioned the brains of postnatal day (P) 0, P3, P7, P10, P14 and P30 C57BL/6 mice for immunohistochemistry. We tracked microglial proliferation by labelling microglia and proliferative cells with IBA1 and KI67 antibodies respectively; a cell positive for both markers indicated proliferating microglia (Fig. 1A,D). KI67 provides a snapshot of cells in the G1, S or G2 phases of the cell cycle (Scholzen and Gerdes, 2000); thus, when combined with IBA1, KI67 labelling is ideal for the study of proliferative microglia at discrete time points across postnatal brain development. It must be noted that IBA1 is expressed in many myeloid cell populations and not restricted to

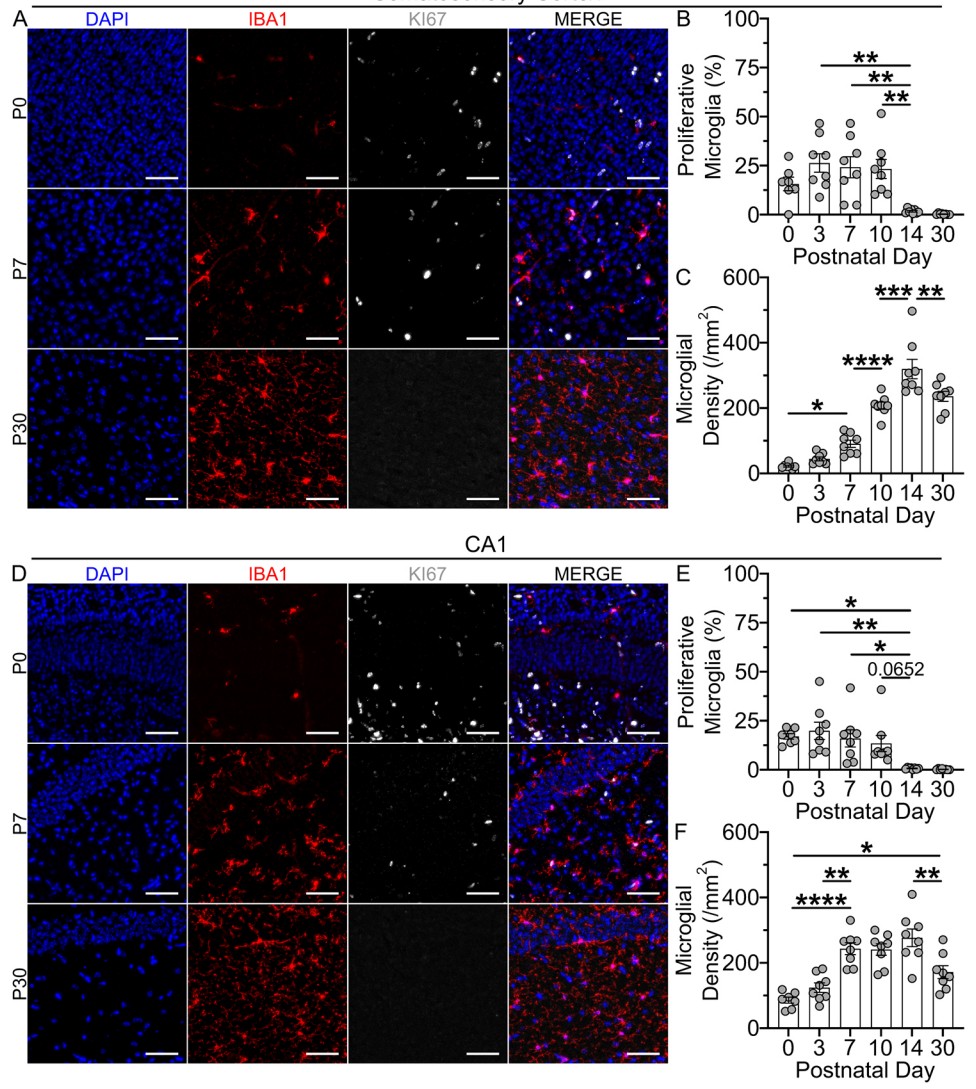

**Fig. 1. Developing microglia proliferate within the first two postnatal weeks.** (A) Representative images of IBA1$^+$ microglia (red) and KI67$^+$ proliferative cells (white) within P0, P7 and P30 somatosensory cortex. Scale bars: 50 μm. (B) Plot of the mean percentage of the cortical microglia that are proliferating at each developmental time point. (C) Plot of cortical microglial densities across development. (D) Representative images of IBA1$^+$ microglia (red) and KI67$^+$ proliferative cells (white) in P0, P7 and P30 CA1 region of the hippocampus. Scale bars: 50 μm. (E) Plot of the mean percentage of CA1 microglia that are proliferating at each developmental time point. (F) Plot of CA1 microglial densities across development. In B, C, E, F, bars represent mean±s.e.m.; $n$=8 mice per time point. Counts of microglia and proliferating microglia were averaged from three or four images in either the somatosensory cortex or CA1. *$P$<0.05, **$P$<0.01, ***$P$<0.001, ****$P$<0.0001 (one-way ANOVA, Tukey post-hoc).

microglia. Thus, we first assessed whether the canonical microglial marker TMEM119 may suitably track developmental microglial proliferation. However, we found minimal expression of TMEM119 between P0 and P10 in both regions, although by P14 approximately 70% of all IBA1 cells in the somatosensory cortex, and approximately 50% of all IBA1 cells in CA1, expressed TMEM119 (Fig. S1A-C). A similar finding has been reported at the transcript level: TMEM119 and other canonical microglial markers were not found at P4/P5 in the developing brain (Hammond et al., 2019). Together, these findings suggest that TMEM119, and potentially other canonical microglial markers, may not be ideal for tracking microglia in early postnatal development. Thus, given the robust IBA1 expression and parenchymal location of these IBA1-labelled cells, we will refer to them as microglia throughout this work.

In the developing somatosensory cortex and CA1 of the hippocampus, microglia proliferated robustly during the first two postnatal weeks, with a cessation of nearly all microglial proliferation by P14 (Fig. 1). In our analyses, approximately 20-25% of all microglia were proliferative at P0, P3, P7 and P10 in the somatosensory cortex, and approximately 20% of all microglia were proliferative in the hippocampus at the same time points (Fig. 1B,E). In both regions, microglial proliferation was minimal by P14 and remained minimal at P30. The early phase of proliferation coincided with a pronounced increase in microglial densities that peaked at P14 in the somatosensory cortex and earlier at P7 in CA1 (Fig. 1C,F). Interestingly, we found microglial densities to decrease in both regions between P14 and P30, which may reflect a brief period of microglial apoptosis after P14, as has previously been reported (Nikodemova et al., 2015). These findings suggest that the timing of microglial proliferation is likely regionally regulated, as is the case for microglial turnover in adulthood (Askew et al., 2017; Tay et al., 2017). Together, an initial phase of postnatal proliferation drives microglia toward a stable density that is reached by P7 in CA1 and by P14 in the somatosensory cortex.

### Several factors are predicted to signal onto microglia at peak developmental proliferation

Growth factors that specifically promote proliferation often do so via paracrine signalling. These factors, called mitogens, are produced and secreted by one cell to act on neighbouring cells that express the mitogen-specific receptor (Zhu and Thompson, 2019). Presumably, the receiving cells then proliferate to achieve a density that permits physiological function that would not otherwise be possible. Given the importance of mitogens in initiating cellular proliferation for other CNS cells, such as oligodendrocyte progenitor cells (Barres and Raff, 1993; Barres et al., 1994; Noble et al., 1988; Raff et al., 1988), we reasoned that CNS cells developing alongside microglia may release microglial mitogens to control their developmental proliferation.

To explore potential cell–cell communication during development, we used CellChat (Jin et al., 2021), a bioinformatic tool that predicts ligand–receptor interactions in scRNAseq datasets, on a publicly available dataset. We searched the Gene Expression Omnibus (GEO) database for an scRNAseq dataset containing as many cell types as possible in the developing brain when microglia are highly proliferative, ideally between P7 and P10. As a secondary criterion, we aimed to find a dataset in which cells were separated based on brain region. With these criteria, we selected the dataset published by Joglekar and colleagues (Joglekar et al., 2021). They prepared an scRNAseq dataset from the prefrontal cortices and hippocampi of P7 mice. We first conducted quality control to exclude doublets or dying cells by analysis of total gene and mitochondrial gene counts, respectively (Fig. S1B) before pooling each of the two cortical

datasets (*n*=3 cortices total) and each of the two hippocampal datasets (*n*=3 hippocampi total) into a single cortical and a single hippocampal dataset. With this, we had 6400 cells in the cortical dataset, of which 409 were microglia, and 14,077 cells in the hippocampal dataset, of which 483 were microglia. We conducted further quality control on each dataset and used Clustree (Zappia and Oshlack, 2018) to determine the appropriate resolution for clustering, and clustered prefrontal cortex and hippocampus datasets from the P7 mouse brain (Fig. S2A-C). In the P7 prefrontal cortex, we identified 20 distinct cell populations and subsets based on differential gene expression patterns. Among these, we identified microglia, brain-associated macrophages (BAM), two oligodendrocyte progenitor cell (OPC) subsets, dividing OPCs (Div OPC), myelinating oligodendrocytes (Myel OL), glioblasts, three astrocyte subsets (Astros 1-3), three excitatory neuron subsets (Excitatory 1-3), inhibitory neurons, Cajal–Retzius cells, fibroblasts, pericytes, endothelial cells, and reticulocytes (Fig. 2A). We performed CellChat to explore potential signalling onto microglia from other cell populations/subsets and filtered the predicted results to 'Secreted Signalling' factors of the CellChat database. We identified 13 probable interactions from 15 of 20 cell populations/subsets that encompassed eight unique ligands signalling onto a suite of nine microglial receptors (Fig. 2B). We identified pleiotrophin (*Ptn*), midkine (*Mdk*), galectin-9 (*Lgals9*), interleukin 34 (*Il34*), fractalkine (*Cx3cl1*), colony-stimulating factor 1 (*Csf1*), chemokine ligand 4 (*Ccl4*) and angiopoietin-like 4 (*Angptl4*) as potential microglial mitogens (Fig. 2B). Likewise, in the P7 hippocampus, we identified 18 cell populations, including those from the prefrontal cortex, as well as ependymal and choroid plexus epithelial cells (CP Epithelial) which were unique to the hippocampus (Fig. 2C). We performed CellChat on this hippocampus dataset and found 17 of 18 cell populations/subsets were predicted to signal onto microglia with a suite of 12 ligands predicted to signal onto 11 microglial receptors. All seven factors identified in the cortex were predicted in the hippocampus; however, transforming growth factor β 1 and 2 (*Tgfb1* and *Tgfb2*), growth arrest-specific 6 (*Gas6*), and chemokine ligand 3 (*Ccl3*) were identified uniquely within the hippocampus (Fig. 2D). While many of these factors have established roles in regulating microglial viability or functional dynamics (Bohlen et al., 2017; Paolicelli et al., 2011; Utz et al., 2020), they may possess unknown roles in regulating developmental proliferation.

### CSF1R signalling boosts microglial proliferation in culture

To explore whether the ligand–receptor pairings predicted to signal onto P7 microglia boost microglial proliferation, we elected to test these factors on serum-free, primary microglia isolated from CD1 mice. Although primary CD1 microglia display subtly elevated inflammatory responses in response to lipopolysaccharide stimulation relative to C57BL/6 counterparts, both *in vitro* and *in vivo* (Nikodemova and Watters, 2011), the greater litter sizes, and thus greater yield of microglia, made them a suitable model for our cell culture experiments. Further, the use of outbred mouse stocks for studying immunological mechanisms has been suggested to be a crucial 'next step' in translating findings from the bench to bedside (Enriquez et al., 2020). While cell culture is limited in its ability to recapitulate the suite of interactions that occur *in vivo*, it provides a valuable, high-throughput tool to assay several potential mechanisms. Additionally, serum-free microglia more faithfully resemble *in vivo* microglia than previous culture methods (Bohlen et al., 2017). We isolated primary mouse microglia from the P5-P7 cortex by immunopanning (Fig. 3A), which yielded highly pure (on average 95.4%) CD11b (encoded by *Itgam*)[+] microglial cultures, of which nearly 100% were also TMEM119[+], by day 7-8 *in*

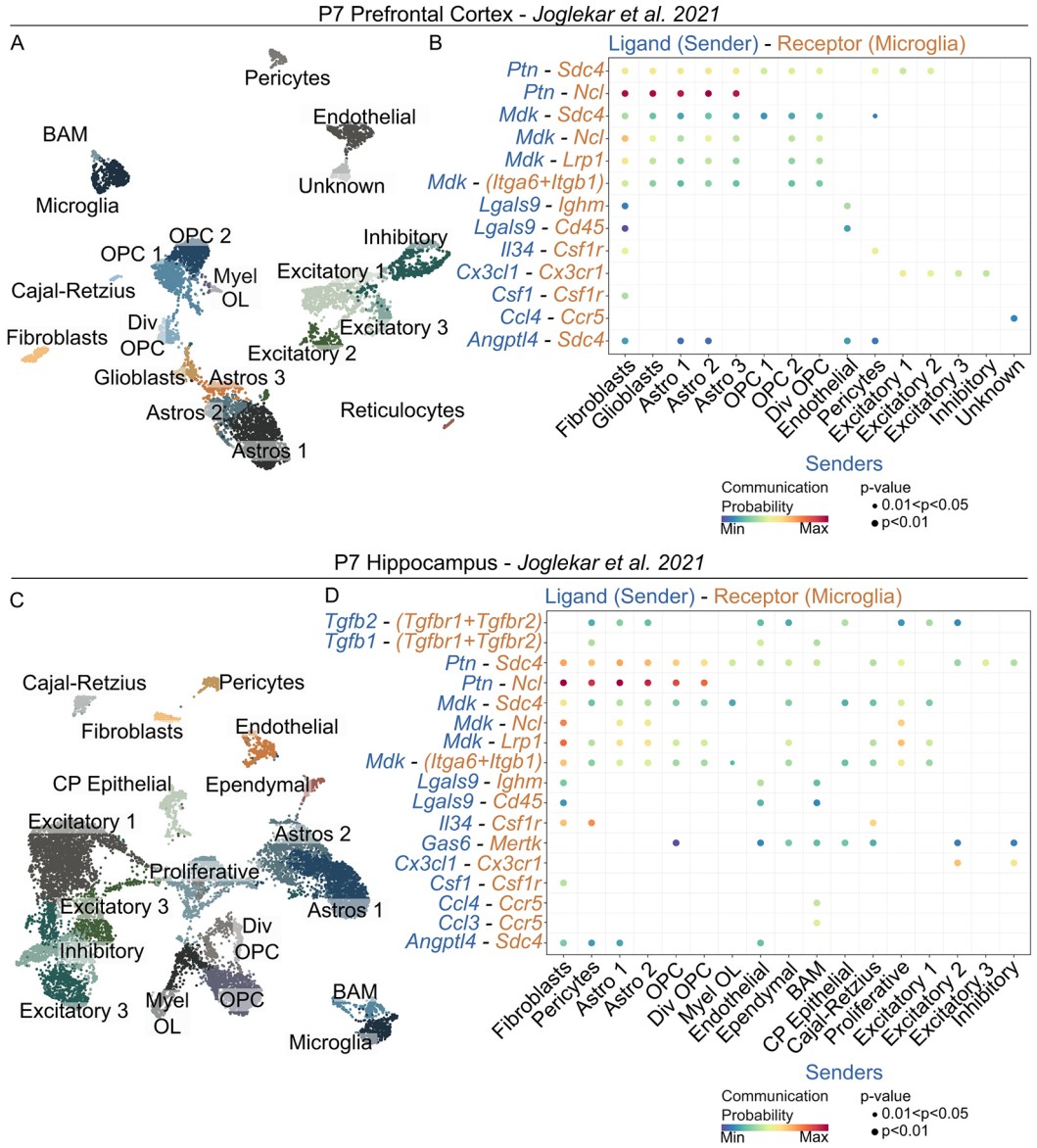

**Fig. 2. CellChat predicts potential ligand–receptor signalling onto microglia in the P7 mouse cortex and hippocampus.** (A) A UMAP plot depicting the cellular populations of the P7 prefrontal cortex using data from Joglekar et al. (2021). Dots represent individual cells clustered based on transcriptional similarity, while colours indicate distinct cell lineages or cellular states based on gene expression patterns. (B) A CellChat plot depicting the ligands (blue) secreted from CNS cell populations predicted to signal onto receptors on microglia (orange) in the P7 mouse prefrontal cortex. (C) A UMAP plot depicting the cellular populations of the P7 hippocampus using data from Joglekar et al. (2021). Dots represent individual cells clustered based on transcriptional similarity, while colours indicate distinct cell lineages or cellular states based on gene expression patterns. (D) A CellChat plot depicting the ligands (blue) secreted from CNS cell populations predicted to signal onto receptors on microglia (orange) in the P7 mouse hippocampus.

*vitro* (Fig. S3A-C). Considering these cells were isolated from brains with low microglial TMEM119 expression (Fig. S1A,B), the inclusion of TGFβ2 in the serum-free media may promote maturation of microglia *in vitro* as it does *in vivo* (Utz et al., 2020). We cultured microglia for 5 days *in vitro* in serum-free conditions to allow microglia to ramify and mature following the immunopanning process before replacing with fresh media containing treatments for an additional 72 h. After 72 h, we either assessed microglial viability with a calceinAM-propidium iodide assay or proliferation following fixation and immunocytochemical labelling. Viability of microglia increased between day (D) 5 and D8 *in vitro*, suggesting that cell cultures can be maintained for this period (Fig. S3D).

We used our cell culture model to test the 12 predicted ligands by quantifying the proportion of microglia that expressed KI67

following treatment. Given that two of these factors – CSF1 and TGFβ2 – are essential survival factors for microglia, we first conducted a viability assessment with calceinAM and propidium iodide on microglia with an identical timeline of 5 days of culture in microglial growth media (GM; contains 10 ng/ml CSF1, 2 ng/ml TGFβ2), followed by 3 days of growth in base media alone (no CSF1 or TGFβ2), or base media with TGFβ2, CSF1, or TGFβ2 and CSF1 (GM). Both TGFβ2 or CSF1 promoted similar levels of microglial viability to microglial growth media containing both CSF1 and TGFβ2 (Fig. S3E). However, given that we wished to assess microglial proliferation of each factor individually, we elected to test all predicted signalling factors in microglial base media (no CSF1 or TGFβ2). We first cultured microglia for 5 days in microglial growth media (with CSF1 and TGFβ2) before

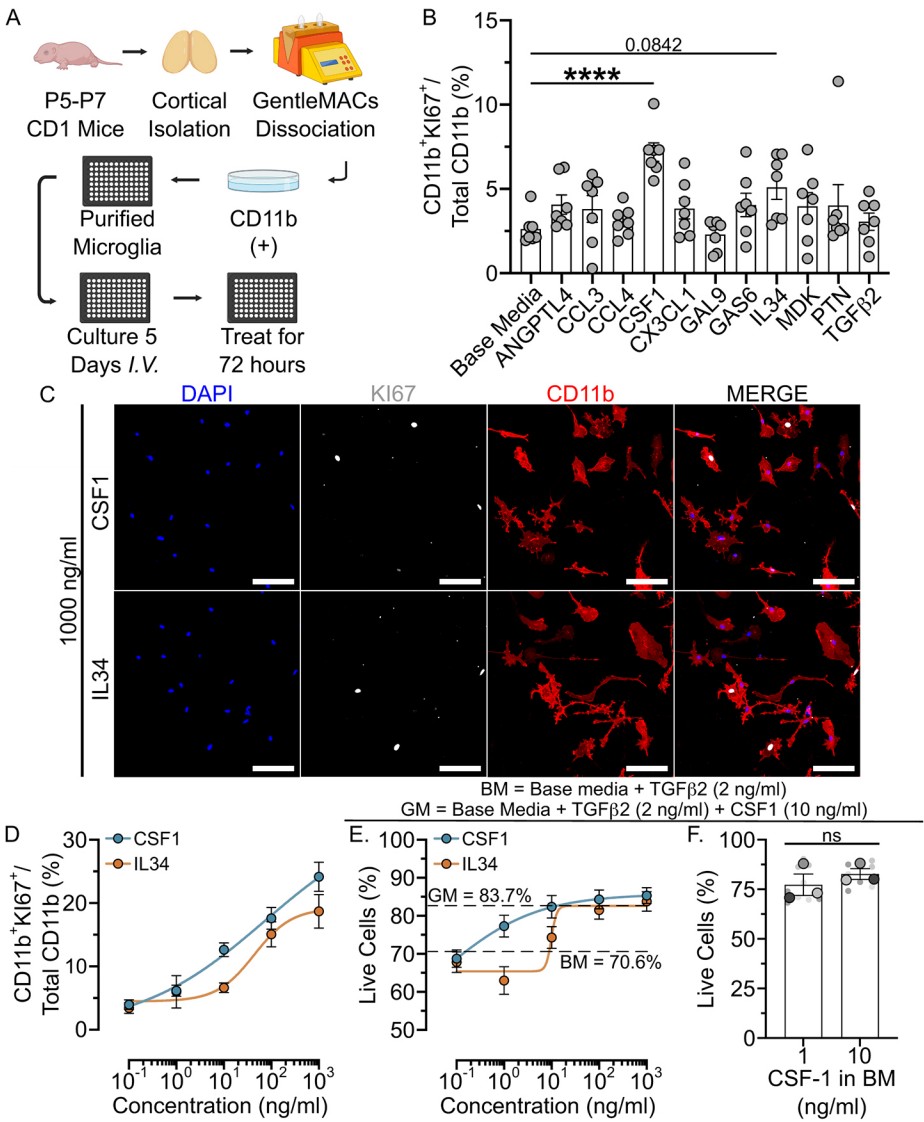

**Fig. 3. CSF1R ligands boost microglial proliferation in serum-free primary microglial culture.** (A) Schematic of immunopanning for the isolation and culture of murine microglia. I. V., *in vitro*. Schematic created in BioRender by Ho, M. 2025. https://BioRender.com/zvx72w4. This figure was sublicensed under CC-BY 4.0 terms. (B) A plot of the percentage of total Cd11b⁺ microglia that were KI67⁺ after a 72-h treatment of 10 ng/ml of each predicted signalling factor in microglial base media. (C) Representative images of proliferative microglia (CD11b⁺KI67⁺) at 1000 ng/ml. Scale bars: 100 µm. (D) A plot of the mean percentage of microglia that are proliferative at tenfold increasing concentrations of CSF1 or IL34. (E) A plot of the mean percentage of viable microglial sustained by tenfold increasing concentrations of CSF1 or IL34. BM here indicates a microglial base media control supplemented with 2 ng/ml TGFβ2; GM indicates microglial growth media control containing both 2 ng/ml TGFβ2 and 10 ng/ml CSF1. (F) A plot of the percentage of live microglia sustained by BM containing either 1 ng/ml or 10 ng/ml CSF1. Bars represent mean±s.e.m.; B: *n*=2, D-F: *n*=3, independent microglial cultures with treatments in triplicate or quadruplicate. ****$P$<0.0001 (one-way ANOVA, Tukey post-hoc). ns, not significant.

performing treatments of each factor at 10 ng/ml in microglial base media for an additional 72 h. Although a lack of both growth factors reduced microglial viability (Fig. S3E), we deemed their viability throughout the treatment period sufficient for these analyses. Of the predicted signalling factors, only CSF1, and IL34 to a lesser extent, boosted microglial proliferation (Fig. 3B).

To ensure that reduced microglial viability did not prevent proliferation induced by non-CSF1R ligands, we also tested each factor in microglia GM and observed no appreciable boost to proliferation (Fig. S4). Both CSF1 and IL34 share a receptor (CSF1R) and so the differences in impacts on proliferation may suggest differential impacts of each at different concentrations. To test this, we conducted dose-response experiments for CSF1 and IL34 (Fig. 3C,D) in a microglial base media that contained only TGFβ2 to enhance survival (Fig. S3C). Both CSF1 and IL34 elicited a robust, dose-dependent boost in proliferative microglia, but CSF1 boosted proliferation to a greater extent than IL34 at lower concentrations (Fig. 3C,D), perhaps owing to differences in binding affinity for CSF1R (Wei et al., 2010). We likewise observed a dose-dependent maintenance of microglial viability by both CSF1 and IL34, but, like proliferation, CSF1 sustained microglial viability to a greater extent than did IL34 (Fig. 3E). Interestingly, microglial

viability was sustained with a dose of 1 ng/ml of CSF1, which did not differ from microglial growth media containing CSF1 at 10 ng/ml (Fig. 3F). Given that 1 ng/ml CSF1 was sufficient to sustain microglial viability (Fig. 3E,F), but a concentration of 10 ng/ml of CSF1 was required to initiate appreciable microglial proliferation (Fig. 3D), CSF1 may regulate distinct aspects of microglial biology at differing concentrations. Alternatively, CSF1 ligands may require a higher concentration to enhance the survival of proliferative cells. While most predicted factors failed to induce microglial proliferation in serum-free culture, these factors may instead regulate other aspects of developmental microglial biology. Alternatively, cultured microglia may downregulate essential receptors that would otherwise be expressed within the CNS. Together, we demonstrated that, of the ligands predicted to signal onto microglia at peak developmental proliferation, CSF1R ligands are highly mitogenic.

## CSF1 and IL34 protein and transcript levels increase throughout development

Considering the capacity for CSF1 and IL34 to boost microglial proliferation in culture, we next assessed the protein and RNA expression for each mitogen throughout development. To assess

protein level changes, we conducted enzyme-linked immunosorbent assays (ELISAs) on whole brain homogenates prepared from C57BL/6 mice at P0, P7, P14 and P30 to track changes on a timeline that encompasses the entirety of microglial development and maturation. Both CSF1 and IL34 increased between P0 and P14, with a distinct peak by P14 (Fig. 4A,B). At P30, IL34 levels, but not CSF1, were reduced. Interestingly, IL34 protein concentrations were roughly 200-750 times greater than CSF1 at all time points.

To map CSF1 and IL34 levels spatially, we used fluorescent *in situ* hybridization (RNAscope™) at P3, P10 and P30, to assess *Csf1* and *Il34* transcript levels throughout the window of developmental microglial proliferation. Specifically, we quantified the proportion of *Csf1*⁺ and *Il34*⁺ cells in CA1 of the hippocampus and in the developmental somatosensory cortex (Fig. 4C). We counted cells as positive if they contained five or more *Csf1* or *Il34* transcripts. In both the hippocampus and cortex, the proportion of *Il34*⁺ cells peaked at P10, was decreased by P30 in the cortex and remained stable between P10 and P30 in the hippocampus (Fig. 4D-G). The proportion of *Csf1*⁺ cells, in contrast to protein levels, remained low and stable in the hippocampus and cortex across development (Fig. 4H-K). The difference in *Csf1* transcript and CSF1 protein levels may reflect substantial regional heterogeneity in CSF1 expression given that RNAscope™ only captures transcript in a small depth of brain,

whereas ELISA of whole brain homogenates captures CSF1 spread throughout the entire brain. We prepared density maps for both *Csf1* and *Il34* that display the transcript density in relation to its nearest neighbour transcript across the entire scan of the brain. Notably, the density of *Il34* transcript increased substantially between P3 and P10 and remained high at P30, as expected. We observed *Il34* hotspots throughout the entirety of the cortex and in the pyramidal layer of the hippocampus (Fig. 4L). In contrast, *Csf1* expression remained consistent throughout development with a more diffuse pattern of expression at P3, although we did observe greater expression in the white matter regions of the corpus callosum and fimbria at P10 and P30, and an elevated level in the granule layer of the dentate gyrus and in the hypothalamus (Fig. 4L). These regionally distinct expression patterns of *Csf1* and *Il34* may reflect differences in their support of local microglia, particularly regarding microglial viability in adulthood when IL34 and CSF1 seem to predominantly support white and grey matter microglial survival, respectively (Badimon et al., 2020; Easley-Neal et al., 2019).

### The *Csf1r* signalling network is restricted to few cell populations in the developing brain

To understand which cell lineages express CSF1R ligands during development, we examined cell-specific expression in the P7

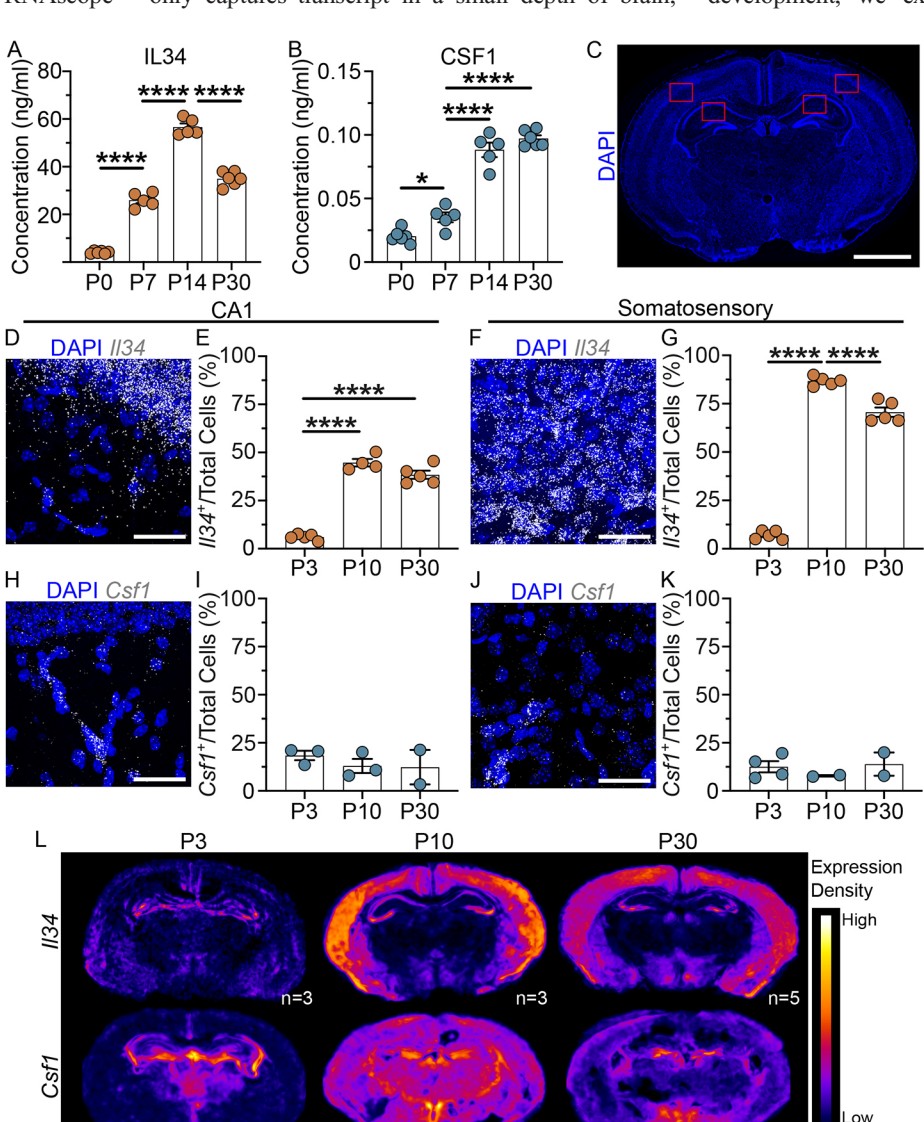

Fig. 4. CSF1 and IL34 protein increase throughout development and display distinct spatiotemporal transcriptional expression patterns. (A,B) IL34 (A) and CSF1 (B) protein levels in whole-brain homogenates as measured by ELISA. (C) Representative brain microscopy image with regions of interest outlined in red. Scale bar: 100 μm. (D,F) Representative images of *Il34* transcript visualized with RNAscope™ in CA1 (D) and somatosensory cortex (F). Scale bars: 40 μm. (E,G) Plots of the mean percentage of total cells of CA1 (E) and somatosensory cortex (G) that are *Il34*⁺ across development. (H,J) Representative images of *Csf1* transcript visualized with RNAscope™ in CA1 (H) and somatosensory cortex (J). Scale bars: 40 μm. (I,K) Plots of the mean percentage of total cells of CA1 (I) and somatosensory cortex (K) that are *Csf1*⁺ across development. (L) Density maps depicting the spatiotemporal expression patterns of *Il34* and *Csf1* transcripts in RNAscope™ across development. Warmer colours indicate greater, while cooler colours indicate lower, transcript density. Bars represent mean±s.e.m.; A,B: n=5 mice per time point; E: n=3-5 mice per time point, G: n=4-5 mice per time point, I,K: n=2-3 mice per time point. *P<0.05, ****P<0.0001 (one-way ANOVA, Tukey post-hoc).

hippocampus scRNAseq dataset (Joglekar et al., 2021). We elected to use the hippocampal dataset given its rich heterogeneity of cell populations and predicted signalling interactions (Figs 2C,D and 5A). Within the P7 hippocampus, microglia and BAM were the sole expressors of *Csf1r* RNA (Fig. 5A,B,E). Interestingly, *Csf1* expression was largely restricted to fibroblasts, with some sparse production in microglia and astrocytes (Fig. 5C,E), whereas *Il34* was enriched in fibroblasts, pericytes and Cajal–Retzius cells (Fig. 5D,E).

As *Csf1r* expression was predominantly limited to microglia and BAM, we explored the likely senders and receivers within the Csf1r signalling network with CellChat (Jin et al., 2021) to further validate the expressors of *Csf1* and *Il34*. For this, CellChat includes both ligands of CSF1R in the predictions for sending populations. Notably, in decreasing order of importance, pericytes, fibroblasts and Cajal–Retzius cells were predicted as the predominant senders of CSF1 and/or IL34 (Fig. 5F). As we anticipated given the highly specific expression of *Csf1r* (Fig. 5E), microglia and BAM were the dominant receivers of CSF1R signalling (Fig. 5F), further supporting the importance of the CSF1R signalling network within the developing brain for these cells.

We reasoned that, if CSF1R signalling were to explicitly regulate the developmental proliferation of microglia, and as both *Csf1* and *Il34* remain high in adulthood as we identified (Fig. 4A,B),

a reduction in CSF1R expression may indicate a reduced sensitivity to both CSF1 and IL34, thus aligning with minimal proliferation in adulthood. To explore this possibility, we determined the expression of *Csf1r* in microglia across a variety of ages, encompassing embryonic (E14), early postnatal (P4/P5), adult (P30, P100) and aging (P540) microglia from another publicly available dataset (Hammond et al., 2019). Contrary to our prediction, *Csf1r* expression increased from E14 to P100, with no drop in expression after P4/5 (Fig. 5G). Together, these data suggest that the CSF1R signalling network is restricted to microglia and BAM. However, *Csf1r* and CSF1R ligand expression levels do not predict the extent of developmental microglial proliferation.

The sparse expression of *Il34* in the scRNAseq dataset used for CellChat analyses contrasted with the robust CA1 *Il34* transcript expression at P10 that we observed with RNAscope. Two factors may explain this discrepancy. First, *Il34* expression may be temporally regulated and dramatically increase between P7 and P10, analogous to the increase between P8 and P14 that occurs in the anterior cingulate cortex, nucleus accumbens and amygdala (Devlin et al., 2024 preprint). Second, scRNAseq may not optimally capture transcriptional data from neurons. Specifically, the fragility and interconnectedness of neurons may render them susceptible to damage/stress artefacts from the enzymatic and mechanical

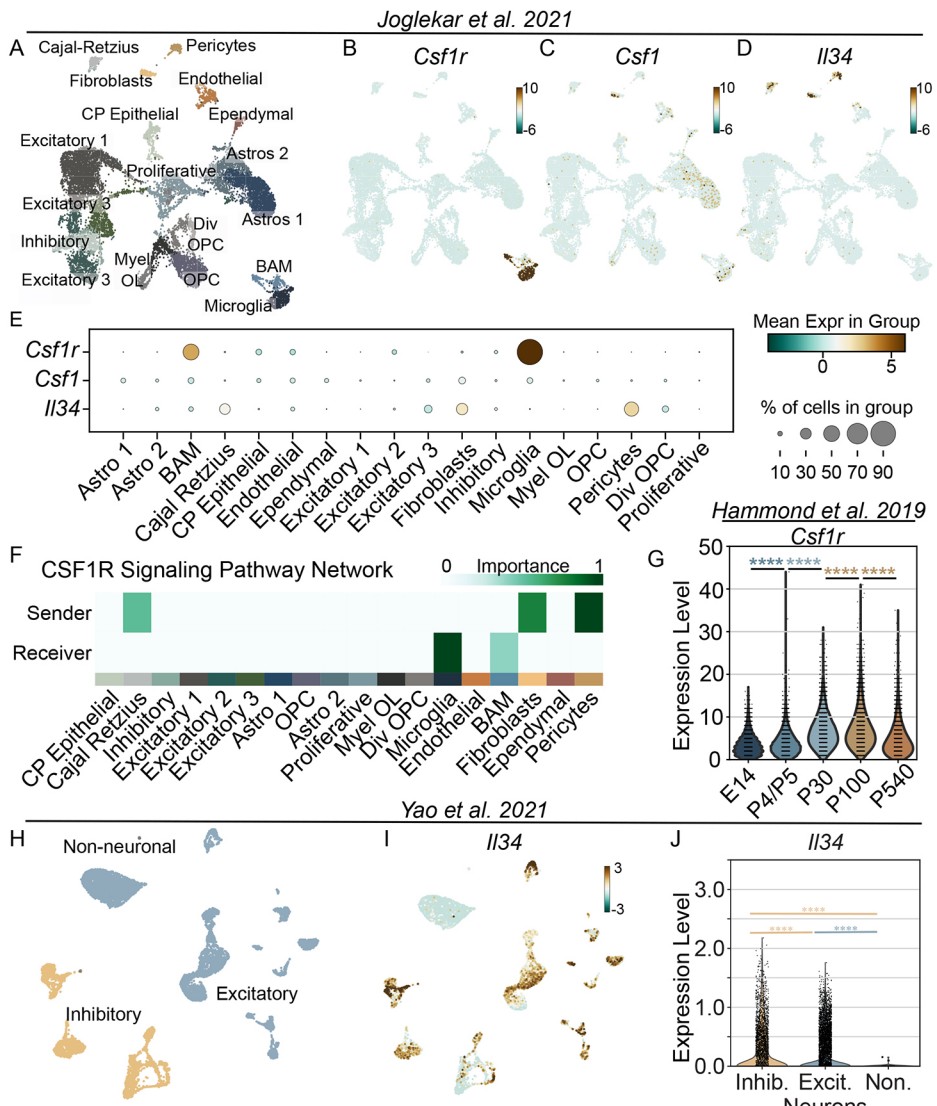

**Fig. 5. CSF1R signalling is largely restricted to microglia and border-associated macrophages in the P7 hippocampus.** (A) UMAP plot depicting the cellular populations of the P7 hippocampus. Dots represent individual cells clustered based on transcriptional similarity, while colours indicate distinct cell lineages or cellular states based on gene expression patterns. (B) UMAP plot depicting expression of *Csf1r* in the P7 hippocampus. (C) UMAP plot depicting expression of *Csf1* in the P7 hippocampus. (D) UMAP plot depicting expression of *Il34* in the P7 hippocampus. (E) Dot plot depicting the cell population expression levels of *Csf1r*, *Csf1* and *Il34* in the P7 hippocampus. (F) CellChat plot predicting the dominant sender populations that secrete CSF1 and IL34 and dominant receiver populations that receive signalling via CSF1R in the P7 hippocampus. (G) Violin plot depicting the expression of microglial *Csf1r* across the mouse lifespan. (H) UMAP plot depicting inhibitory neurons, excitatory neurons and non-neuronal cells of the adult hippocampus. Dots represent individual cells clustered based on transcriptional similarity, while colours indicate distinct cell lineages or cellular states based on gene expression patterns. (I) UMAP plot depicting expression of *Il34* in the adult hippocampus. (J) Violin plot comparing the expression level of *Il34* between inhibitory neurons, excitatory neurons and non-neuronal cells of the adult hippocampus. Datasets for A-F: Joglekar et al., 2021; G: Hammond et al., 2019; H-J: Yao et al., 2021. G,J: ****$P<0.0001$ (one-way ANOVA, Tukey post-hoc).

dissociation techniques used (Joglekar et al., 2021). Great care must be taken in neuronal isolation to avoid such damage/stress (Cuevas-Diaz Duran et al., 2022). Thus, we assessed transcriptional profiles of adult (>P53) mouse neurons from a preparation optimized for neuronal viability and survival (Yao et al., 2021). Neurons were isolated from the hippocampus (HIP); parasubiculum, postsubiculum, presubiculum (PAR-POST-PRE); and subiculum and prosubiculum (SUB-ProS) (Fig. S5A). We clustered neurons and broadly identified excitatory and inhibitory neurons based on *Slc17a7* and *Gad1* expression, respectively (Fig. 5H, Fig. S5B,C), and found that both express robust levels of *Il34*, with greater levels found in inhibitory populations (Fig. 5I,J) of the adult hippocampus.

### *Csf1* and *Il34* expression levels do not fluctuate with aberrant microglial proliferation in development

Considering the overlap of microglial proliferation with the increasing expression of *Csf1* and *Il34* throughout development, and the ability of both factors to boost proliferation in culture, we next aimed to explore whether the expression of these mitogens is altered alongside aberrant developmental microglial proliferation. CSF1R 'loss-of-function' models have developmental abnormalities and lack microglia (Hammond et al., 2021). CSF1 and IL34 are essential for microglial survival and, therefore,

it is challenging to disentangle their proliferative and survival-promoting roles *in vivo*. Fortunately, microglial proliferation has been more widely explored in models of adulthood repopulation following microglial depletion, whereby the depletion of microglia initiates a robust, CNS-wide proliferative phase, analogous to development, that permits microglia to repopulate the entire CNS (Bruttger et al., 2015; Elmore et al., 2014; Huang et al., 2018). Notably, interleukin 1 receptor 1 signalling partially regulates this proliferation following microglial depletion (Bruttger et al., 2015). Given the similarity of microglial proliferation between repopulation models and development, we reasoned that interleukin 1 signalling may similarly regulate developmental microglial proliferation.

To explore this possibility, we first examined microglial proliferation across development at P3, P10 and P30 in the hippocampus and somatosensory cortex of C57BL/6, IL1α knockout and IL1β knockout mice (Fig. 6A,B). Interestingly, microglial proliferation at P3 and P10 in both the hippocampus and somatosensory cortex was significantly greater in both IL1α and IL1β knockouts relative to C57BL/6 controls, but was minimal by P30 (Fig. 6C,D). These altered proliferation dynamics also coincided with altered microglial densities. In both regions of IL1α and IL1β knockouts, microglial densities were lower at P10 but reached a similar density to C57BL/6 controls by P30 (Fig. 6E,F). This simultaneous heightened microglial proliferation and lowered

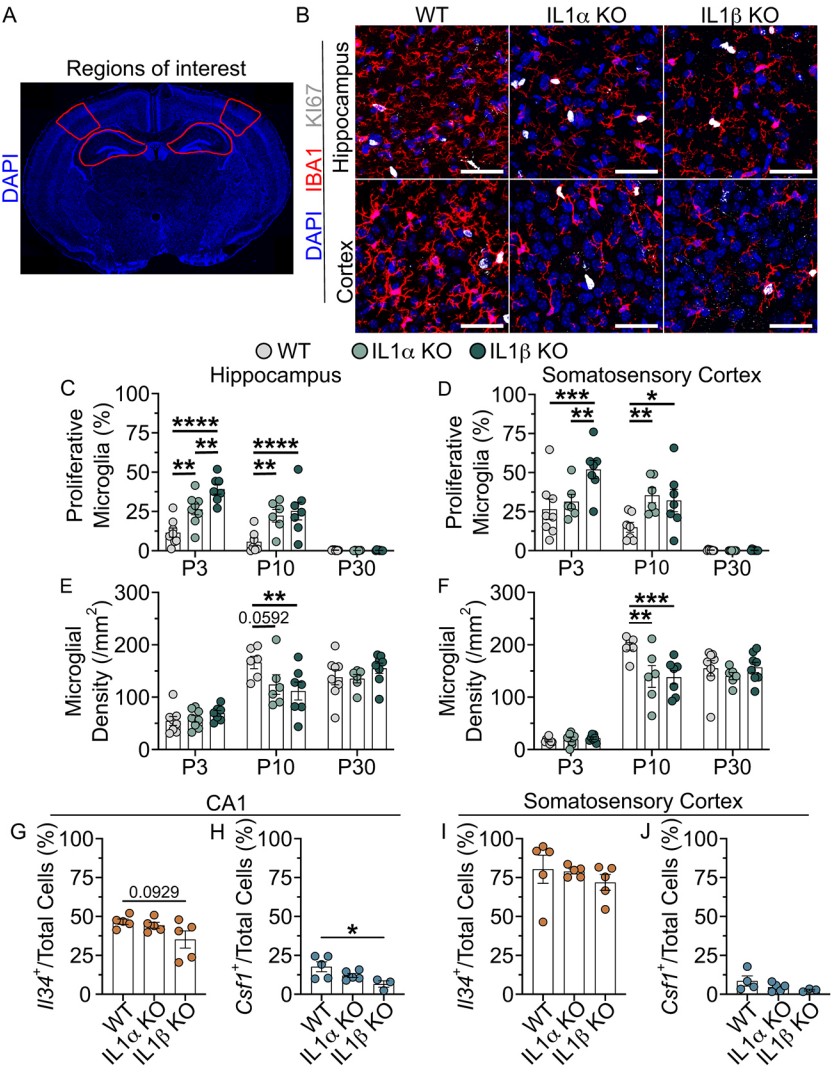

**Fig. 6. Unaltered *Il34* and *Csf1* levels of IL1α and IL1β knockout mice are insufficient to explain aberrant developmental microglial proliferation.**
(A) Representative whole brain microscopy image with analysed regions of interest outlined in red.
(B) Representative images of IBA1⁺ microglia (red) and KI67⁺ proliferative cells (white) in the P10 hippocampus and cortex of IL1α and IL1β knockout mice. Scale bars: 40 μm. (C,D) Plots of the mean percentage of hippocampal microglia (C) and somatosensory cortex microglia (D) that are proliferating in wild-type (WT), IL1α and IL1β knockout mice across development. (E,F) Plots of mean hippocampal microglial densities (E) and mean somatosensory cortex microglial densities (F) in WT, IL1α and IL1β knockout mice. (G,H) Plots of the mean percentage of total cells of CA1 that are *Il34*⁺ (G) or *Csf1*⁺ (H) at P10 in WT, IL1α and IL1β knockout mice. (I,J) Plots of the mean percentage of total cells of somatosensory cortex that are *Il34*⁺ (I) or *Csf1*⁺ (J) at P10 in WT, IL1α and IL1β knockout mice. Bars represent mean±s.e.m.; C-F: n=6-8 mice per time point; G,I: n=5 mice per genotype; H,J: n=3-5 mice per genotype. *P<0.05, **P<0.01, ***P<0.001, ****P<0.0001 (C-F: two-way ANOVA, Tukey post-hoc; G-J: one-way ANOVA, Tukey post-hoc).

microglial density in IL1α and IL1β knockouts may suggest a delay in microglial developmental proliferation in the knockouts.

As IL1α and IL1β knockout mice display aberrant developmental proliferation dynamics relative to C57BL/6 controls, we next explored whether altered *Csf1* or *Il34* expression could explain the heightened microglial proliferation in IL1α and IL1β knockout mice. To this end, we assessed *Csf1* and *Il34* transcript expression at P10 – a time point of prominently altered microglial proliferation – in C57BL/6, IL1α and IL1β knockout brains. Despite higher levels of microglial proliferation in IL1α and IL1β knockout brains, we found no corresponding increase in *Il34* or *Csf1* expression in CA1 (Fig. 6G,H). Contrary to our prediction, the proportions of *Il34*- and *Csf1*-expressing cells were lower in CA1 of IL1β knockouts (Fig. 6G,H), but not in the somatosensory cortex (Fig. 6I,J). Therefore, elevated microglial proliferation in IL1α and IL1β knockout mice is unlikely to be caused by altered *Il34* or *Csf1* expression. We conclude that microglial proliferation is, therefore, regulated by other unidentified mechanisms.

## DISCUSSION

The developmental proliferation of microglia is crucial for the appropriate development of the CNS. Here, we demonstrate histologically that murine microglia proliferate for the first two postnatal weeks, with a peak at P7, to establish a homeostatic density that is maintained into adulthood, thus expanding upon previous assessments of developmental microglial proliferation by flow cytometry (Ginhoux et al., 2010; Nikodemova et al., 2015) and histological assessments of microglial densities (Kim et al., 2015). In testing factors predicted to signal onto microglia at the peak of developmental proliferation, we found that the CSF1R ligands CSF1 and IL34 boost proliferation in primary, serum-free microglia. Although CSF1R ligand-induced proliferation has been described previously, to the best of our knowledge we are the first to disentangle the role of CSF1R ligands in survival and proliferation in serum-free conditions that generate microglia similar to *in vivo*, developmental conditions (Bohlen et al., 2017). Many of the remaining factors that we identified in our receptor–ligand analysis of developing microglia have no known impact on microglia, which raises intriguing questions regarding their potential developmental functions. We also found that both CSF1 and IL34 increase throughout development in distinct spatial and temporal patterns, as has been recently described (Devlin et al., 2024 preprint), and that IL1α and IL1β knockout mice have abnormal developmental microglial proliferation, which was not associated with compensatory alterations in *Csf1* and *Il34* transcript expression. These findings support the idea that, while microglia are capable of robust proliferation at distinct developmental time points and in response to highly specific mitogenic signalling, at least one additional regulatory factor of developmental proliferation remains to be discovered.

It may initially seem surprising that, out of 12 ligands predicted to signal onto microglia at the peak of developmental proliferation, only two, let alone two that share a common receptor, directly boost microglial proliferation. However, CellChat predicts any potential ligand–receptor interactions, not solely those that may impact proliferation. It is entirely possible that the predicted non-mitogenic interactions regulate other developmental microglial dynamics, such as recruitment or migration, within the developing CNS. Such non-mitogenic functions may indirectly facilitate developmental microglial proliferation. For example, following microglial depletion in *Cx3cr1* knockout mice, which renders microglia incapable of responding to fractalkine (CX3CL1), the spontaneous repopulation of the retina is temporarily delayed

(Zhang et al., 2018). Conversely, microglial repopulation occurs more rapidly following exogenous administration of CX3CL1 during repopulation.

Outside of repopulation, CX3CR1 deficiency similarly delays the acquisition of the adulthood microglial density in development (Paolicelli et al., 2011). This may, at first glance, appear to conflict with our finding that CX3CL1 is not mitogenic; however, CX3CL1, rather than directly inducing microglial proliferation, might instead promote microglia migration within the developing CNS. In corroboration of this, CX3CL1 accelerates microglial migration in *in vitro* chemotaxis assays (Zhang et al., 2012). Such signalling may ensure that microglia are present and able to proliferate in regions where their developmental functions, such as phagocytosis, are more immediately required. In support of this, CX3CL1 may exert local control over developmental microglial biology whereby microglial CX3CR1 signalling induces the release of WNT onto astrocytes, which initiates astrocytic pullback from synapses and subsequent synaptic engulfment by microglia (Faust et al., 2024 preprint).

As CX3CR1 deficiency does not yield a permanent reduction in microglial densities, this may suggest redundancy in microglial recruitment and chemotactic mechanisms. For example, the heparin-binding growth factors pleiotrophin (PTN) and midkine (MDK), which share 50% sequence homology and are robustly expressed in murine development and were predicted to display strong communication probability onto microglia in our analyses, have been minimally explored for their signalling functions in microglia; their capacity to promote recruitment and migration of other myeloid lineages may suggest a similar function in microglia and their progenitors (Sorrelle et al., 2017). Likewise, the lack of mitogenicity of CCL3 and CCL4, despite the temporally restricted expression of their shared receptor, CCR5, in developing rat microglia (Cowell et al., 2006), may suggest an unidentified chemotactic capacity. Thus, while many of the ligands we explored failed to elicit microglial proliferation, each may exert indirect control over developmental microglial proliferation, although the precise mechanisms remain to be elucidated. Additionally, although we selected an scRNAseq dataset within the timeline of developmental microglial proliferation for our CellChat analyses, we cannot discount the possibility that transient progenitor cell populations or cellular states that disappear prior to P7, as has been described throughout embryogenesis (La Manno et al., 2021), initiate microglial proliferation, and we simply observed the tail-end of this interaction at P7. Similarly, it may be that cell populations not captured in the scRNAseq dataset preparation, due to poor viability or quality control of such cells, could influence microglial proliferation.

Here, we did not identify a definitive microglial mitogen with an expression pattern that mirrors developmental proliferation. Although both CSF1 and IL34 are potent microglial mitogens in *in vitro* assays, and both increase throughout brain development, the fact that their expression levels remain high well into adulthood, when microglial proliferation is minimal, implies that they are not solely responsible for developmental microglial proliferation. While we do not discount the possibility that an unknown mitogen may regulate developmental proliferation, we propose that an additional, underappreciated layer of regulation may exist in the contact-mediated inhibition of proliferation once an adulthood density is reached. A recent study elegantly demonstrated that developing microglia clonally proliferate by allometric scaling whereby proliferation is limited by the physical space available within the brain (Barry-Carroll et al., 2023). Once microglia reach their maximal density, the continual extension and retraction of microglial processes facilitates surveillance of the CNS whereby

microglia–microglia contacts result in rapid retraction of microglial processes such that microglia only survey a distinct spatial domain (Hines et al., 2009; Nimmerjahn et al., 2005). Such microglia–microglia, or even microglia–glia or microglia–neuron, contacts may simultaneously limit microglial proliferation. However, the mechanisms of this proposed contact inhibition in microglia are unknown.

CSF1R signalling crucially supports microglial survival (Badimon et al., 2020; Bohlen et al., 2017; Elmore et al., 2014; Hammond et al., 2021). In our primary microglial culture, we confirmed a direct role for both CSF1 and IL34 in promoting both microglial viability and proliferation. However, conducting suitable loss-of-function experiments to explore how each ligand differentially impacts viability or proliferation, either *in vitro* or *in vivo*, complicates assessments of each independently of the other. Despite this shortcoming, the fact that we found that CSF1 facilitated both survival and proliferation, whereas TGFβ2 facilitated survival independent of proliferation, lends support to the idea that CSF1 can promote proliferation independently of survival. A recent study likewise suggests a role for microglial proliferation that is independent of survival; IL34 knockout reduces cortical microglial numbers at P8 and P15, although it remains unclear whether there is simultaneously elevated microglial death as a result of insufficient CSF1R signalling (Devlin et al., 2024 preprint).

Suitable models for mechanistically exploring aberrant developmental microglial proliferation remain elusive. Here, we used IL1α and IL1β knockout mice as a model of abnormal proliferation, and, although we can adequately assess changes in proliferation, this remains an imperfect model given a lack of interleukin 1 receptor 1 (IL1R1) expression in microglia (Liu et al., 2019). Thus, any impacts on microglial development are presumably mediated by an additional cell lineage, such as astrocytes, endothelial cells, or neurons. For example, endothelial and epithelial IL1R1 recruit other immune cells to the CNS in non-developmental contexts; IL1α and/or IL1β may likewise recruit microglial progenitors to the developing CNS (Ching et al., 2007; Li et al., 2011). A lack of either cytokine may delay microglial infiltration and subsequently augment microglial proliferation within the developing CNS. Thus, we cannot discount the possibility that, rather than stemming from altered mitogen levels, the elevated developmental microglial proliferation in the IL1α and IL1β knockout mice may relate to delayed allometric expansion of microglia within the brain; however, we are unable to provide evidence of this in our current study.

Overall, by exploring the developmental proliferation of microglia, we confirmed that microglial proliferation is temporally restricted to the first two postnatal weeks. While we identified two likely candidate mitogens that may promote developmental proliferation, the strict timeframe of proliferation may not be a direct result of mitogen expression levels. Given these findings, there is likely an additional layer of complexity to the regulation of developmental microglial proliferation that must be elucidated in future studies.

## MATERIALS AND METHODS
### Animals
For the developmental timeline of microglial proliferation, we obtained pregnant C57BL/6 mice from Charles River Canada and collected progeny at the following ages: P0 (*n*=7), P3 (*n*=8), P7 (*n*=8), P10 (*n*=8), P14 (*n*=8) and P30 (*n*=8), for a total of 47 mice. For the developmental timeline of microglial proliferation in C57BL/6, IL1α and IL1β knockouts, we collected six to eight mice per genotype at each age assessed (P3, P10 and P30). For cell culture experiments, we obtained pregnant CD1 mice (E13.5-E15.5) from Charles River Canada. For histology and culture experiments, we used mice of both sexes given difficulties in determining the sex of mice prior to

P10. All animal experiments and procedures used were reviewed and approved by the animal ethics committees at Université Laval and the University of Alberta.

### Histology
#### Tissue processing
For histology, we euthanized C57BL/6 mice at experimental endpoints by decapitation for P0, P3 and P7 mice or euthanasia by intraperitoneal Euthanyl injection (sodium pentobarbital; Vetoquinol, 127819) for P10, P14 and P30 mice. We transcardially perfused mice older than P7 with 20 ml of room temperature 1× PBS followed immediately by perfusion with 20 ml of ice cold 4% paraformaldehyde (Millipore-Sigma, 441244-3KG) in 0.1 M phosphate buffer [sodium phosphate dibasic anhydrous (Fisher Scientific, 374-1) and sodium dihydrogen phosphate monohydrate (Anachemia, 10049-21-5) in water]. Brains were removed and then submerged in 4% paraformaldehyde for up to 24 h at 4°C to post-fix brains. Brains were then submerged in 30% sucrose [D-sucrose (Fisher Scientific, BP220-1) in 0.1 M phosphate buffer] for up to 72 h at 4°C to cryopreserve the tissue. The tissue was then surrounded in optimal cutting temperature compound (Fisher Scientific, 14-373-65) and snap-frozen over liquid nitrogen for extended storage at −80°C until sectioning. To section the tissue, we coronally cryosectioned brains at 20 μm thickness at −20°C for the entirety of the hippocampus using a Leica CM1950 cryostat. Sections were adhered onto slides (Thermo Fisher Superfrost Slides, 22-037-246) and stored at −80°C until used for histology.

#### Immunohistochemistry
To track developmental microglial proliferation, we performed fluorescence immunohistochemistry. Briefly, we air-dried sections for 30 min, rehydrated in PBS for 10 min and performed antigen retrieval in a citric acid buffer (pH 6.0) composed of anhydrous citric acid (Sigma-Aldrich, 251275), 0.5% Tween-20 (Fisher Scientific, BP337-500) and ddH$_2$O at 60°C for 10 min. We washed slides in PBS and blocked for 45 min with a solution of 10% normal donkey serum (Millipore-Sigma, D9663), 0.1% gelatine from cold water fish skin (Truin Science, FG800), 0.1% Triton X-100 (Fisher Scientific, BP151-500) and 0.05% Tween-20 (Fisher Scientific, BP337-500) in 0.01 M PBS. We diluted the primary antibodies – rabbit anti-IBA1 (1:1000; Wako, 019-19741), goat anti-IBA1 (1:400; Novus Biologicals NB100-1028), rabbit anti-Tmem119 (1:400; Abcam, ab209064) and rat anti-KI67 (1:200; Invitrogen, 14-5698-82) in a buffer of 0.1% gelatine from cold water fish skin and 0.1% Triton X-100 in 0.01 M PBS and incubated slides with primary antibodies at 4°C overnight. We washed slides three to six times with PBS and incubated slides with secondary antibodies anti-rabbit Alexa Fluor 488 (1:400; Jackson ImmunoResearch, 711-546-152), anti-goat Alexa Fluor 488 (1:400; Jackson ImmunoResearch, 705-546-147), anti-rat Cy3 (1:400; Jackson ImmunoResearch, 712-166-153), anti-rat Alexa Fluor 594 (1:400; Jackson ImmunoResearch, 712-586-153), anti-rabbit Alexa Fluor 647 (1:400; Jackson ImmunoResearch, 711-606-152) or anti-rat Alexa Fluor 647 (1:400; Jackson ImmunoResearch, 712-606-153) conjugated *F*(ab)2 fragments antibodies for 1-2 h at room temperature. We included DAPI (1:1000; Invitrogen, D1306) in the secondary antibody incubation to label nuclei. We washed off secondary antibodies with three to six washes of PBS and mounted coverslips (Fisher Scientific, 12541033CA) on slides with Fluoromount G (Invitrogen, 00-4958-02). We acquired *z*-stacks within CA1 and the somatosensory cortex at 20× magnification on a Leica Thunder Imager epifluorescence microscope. For the IL1α and IL1β knockout timeline analyses, we acquired *z*-stacks at 20× magnification on a Zeiss Axioscan.Z1, and for representative images we acquired *z*-stacks at 40× on a Leica Stellaris 8 (University of Alberta Cell Imaging Core).

#### RNAscope™
To visualize the developmental expression of *Csf1* and *Il34*, we performed RNAscope as per the ACDbio Multiplex V2 protocol (Doc. No. UM323100) for fixed frozen tissue. All incubation steps were carried out inside a HybEZ II Hybridization oven (ACD, 321710-R). We conducted 60°C incubations in a dehumidified chamber or 40°C incubations in a humidified chamber with the humidity control tray (ACD, 310012) and

EZ-Batch Slide Holder (ACD, 321716). Briefly, we rehydrated slide in PBS for 5 min, incubated slides at 60°C for 30 min, then post-fixed in fresh 4% paraformaldehyde for 30 min at 4°C. Next, we dehydrated tissue in a series of 50%, 70%, 100% ethanol (twice) (Greenfield Global, P016EAAN), air-dried for 5 min, treated with hydrogen peroxide from the ACD H2O2 and Protease Reagents kit (ACD, 322381) for 10 min at room temperature followed by target retrieval in RNAscope target retrieval reagent (ACD, 322000) at 100°C for 5 min before a wash in distilled water and a final dehydration in 100% ethanol drying step at 60°C for 5 min. Once air-dried, we incubated tissue with a mild protease (RNAscope™ Protease Plus; ACD, 322381) at 40°C for 15 min and washed in distilled water. For the remainder of the protocol, all wash steps were performed in RNAscope wash buffer (ACD, 310091) twice for 2 min per wash. Next, we hybridized tissue with either a *Csf1* (ACD, 315621-C2) or *Il34* (ACD, 428201) mRNA probe for 2 h at 40°C. After washing, we amplified the signal via a series of amplification reagents – AMP1, AMP2 and AMP3 at 40°C for 30, 30 and 15 min each respectively – with a wash between each reagent. We developed the signal for each mRNA probe as per the manufacturer's protocol, with washing between each step. We first added a horseradish peroxidase for 15 min at 40°C probe to permit binding of a fluorescent probe. We incubated tissue for 30 min at 40°C with either Opal 570 (Akoya Biosciences, FP1488001KT) or Opal 650 (Akoya Biosciences, FP1496001KT) to tag the mRNA probes and prevented further probe binding with a horseradish peroxidase blocker. We developed the next channel in the same fashion until either or both probes were labelled. All signal amplification, signal detection and counterstaining was performed with reagents from the RNAscope Multiplex Fluorescent Detection Reagents v2 kit (ACD, 323110). For analyses, we acquired z-stacks at 20× on a Zeiss Axioscan.Z1, and for representative images we acquired z-stacks at 40× on a Leica Stellaris 8 (University of Alberta Cell Imaging Core).

### ELISA

For ELISA, we euthanized C57BL/6 mice at P0 ($n=5$) and P7 ($n=5$) by decapitation and at P14 ($n=5$) and P30 ($n=5$) by euthanasia by intraperitoneal injection of Euthanyl (sodium pentobarbital; Vetoquinol, 127819). P14 and P60 mice were perfused with 20 ml of room temperature PBS. We dissected brains, removed the olfactory bulbs and cerebellum and mechanically dissociated the tissue using a Dounce homogenizer in a solution of tissue extraction reagent (Invitrogen, FNN0071), phosSTOP (Roche, 04906837001) and cOmplete EDTA protease inhibitor cocktail (Millipore-Sigma, 11873580001) and spun the samples at 10,000 *g* for 10 min at 4°C. Supernatants were collected and then stored at −80°C until use.

Prior to use in the ELISA, we conducted a Bio-Rad DC protein estimation assay (Bio-Rad, 500-0116). Briefly, we diluted 5 µl of each sample with 25 µl of Reagent A and 200 µl of Reagent B, incubated at room temperature for 15 min, and read absorbances at 750 nm on a Bio-Rad xMark™ Microplate Spectrophotometer. We determined sample concentrations by plotting sample absorbances on a 4-parameter logistic curve generated from the absorbances of the protein standards.

Both IL34 (R&D Systems, M3400) and CSF1 (R&D Systems, MMC00B) ELISAs were conducted as per the manufacturer's instructions. Briefly, we normalized all samples to 200 µg of total protein, performed the ELISA and read absorbances at 450 nm with a 540 nm wavelength correction. To determine total concentrations in the brain homogenates, we multiplied the resultant concentration by the dilution factor required to normalize samples at 200 µg total protein prior to conducting the ELISA.

### Bioinformatics
#### Quality control and clustering

We downloaded a publicly available dataset (Joglekar et al., 2021; GSE158450) to probe potential ligand–receptor interactions between other brain cell lineages and microglia in the developing prefrontal cortex and hippocampus (P7). We conducted the initial quality control, dimensionality reduction and early clustering using Seurat (Hao et al., 2021; Stuart et al., 2019) (v.5.0; https://github.com/satijalab/seurat) in R. For each of the two prefrontal cortex and hippocampal replicates, we created a Seurat object that included genes expressed in at least three cells and cells expressing at least 200 genes with the CreateSeuratObject()

function, with further refinement to exclude cells with high gene counts (>3000 genes) and dead cells by removing high percentages of mitochondrial genes (>10%), as these are likely doublets or dead cells, respectively. We merged the two prefrontal cortex replicates and two hippocampal replicates using the merge() function and normalized using the SCTransform() function to normalize according to a binomial regression model (Hafemeister and Satija, 2019). Within each merged object, we integrated each dataset with Harmony to reduce batch effects between samples.

We performed the initial dimensionality reduction using RunPCA(), FindNeighbors() (Dimensions=15) and FindClusters() functions. We used 15 principal components for downstream analyses. We ran the FindClusters() function at all resolutions between 0.1 and 1.0 in 0.1 increments and used clustree() from the Clustree package (Zappia and Oshlack, 2018) and determined that a resolution of 0.5 was the most stable clustering resolution.

To refine the clustering of each dataset further, we applied the Single Cell Clustering Analysis Framework (SCCAF v.0.0.10; https://github.com/SCCAF/sccaf) in a Jupyter notebook (v.6.0.3) running a Python environment (Python v.3.8.3). To do so, we converted each SeuratObject to an h5Seurat file, which we then converted into an h5ad file that could then be read into a Jupyter Notebook in the Python environment using Scanpy (v.1.6.0). We used the SCCAF_optimize_all() function (minimum accuracy=95%, iterations=150) on each dataset to a 95% self-projection accuracy; this final clustering was then plotted onto a uniform manifold approximation and projection (UMAP). We saved the new clustering information as a csv file, which we then read back into the R statistical environment to apply the final clustering iteration to the original SeuratObject of both the prefrontal cortex and the hippocampus.

To create a dot plot to explore gene expression across different cell types of the hippocampus, we converted the P7 hippocampus SeuratObject to an h5ad file using SeuratDisk by first converting the RDS file to an h5Seurat file using the Saveh5Seurat() and converting the h5Seurat file to an h5ad file using the convert() function (Destination=h5ad). The h5ad file was read into the Python environment using Scanpy (v.1.6.0) and we created a dot plot using pl.dotplot().

To track microglial *Csf1r* expression across the mouse lifespan, we loaded the annotated dataset as the operating Ident (Ident="Paper_Cluster") from the NIH GEO database (GSE121654) as published by the original authors (Hammond et al., 2019) and removed the monocytes and macrophages using the subset() function (Idents="Mono/Mac", invert=TRUE). We further reduced the dataset to include only cells collected from mice aged E4, P4/5, P30, P100 and P540 using the subset() function [Idents=c ("E4", "P4/5", "P30", "P100", "P540")]. All age groups were downsampled to include 6500 cells using the subset() function (downsample=6500) and the *Csf1r* gene was plotted across all age groups on a violin plot using the VlnPlot() function. We compared all genes between each permutation of age group pairings using the FindMarkers() function on the RNA assay (only.pos=FALSE, min.pct=0, logfc.threshold=0, slot="data", assay="RNA").

To assess whether neurons were enriched in *Il34* in the adult brain, we examined an scRNAseq dataset of hippocampal brain regions via computational analysis in R using Seurat (v.4.4). To achieve this, we used previously published scRNAseq data (Yao et al., 2021) from a Smart-seq dataset and associated metadata of the cortex and hippocampal formation were downloaded. Cells were from the hippocampal region (HIP); parasubiculum, postsubiculum, presubiculum (PAR-POST-PRE); and subiculum and prosubiculum (SUB-ProS) regions were extracted and used for analysis. We created a Seurat object using CreateSeuratObject (min.cells=3, min.features=200) and normalized gene expression in these cells (*n*=9757) via NormalizeData(). We assessed variable genes via FindVariableFeatures(selection.method="vst", nfeatures=2000). We then ran ScaleData(), RunPCA(), FindNeighbours(), FindClusters(resolution=0.15) and RunUMAP() using 20 dimensions, with all other parameters kept at default values, and finally DimPlot() to plot cluster identities, class or region information.

### CellChat

To explore potential signalling onto microglia at P7, we used the CellChat package (v.1.6.1; https://github.com/sqjin/CellChat) to create a CellChat

object using the createCellChat() function for the SCCAF clustering iterations for each of the cortex and hippocampus objects. Within these objects, we identified overexpressed genes and interactions using the identifyOverExpressedGenes() and identifyOverExpressedInteractions(), respectively, and determined the communication probability with the computeCommunProb(type="truncated mean, trim=0.1). We then predicted the potential communicating pathways using computeCommunProbPathway() and filtered the results to exclude cell populations with fewer than 50 cells using filterCommunication(). We then plotted the predicted signalling pathways onto microglia using netVisual_bubble. To determine the contributions of different cell populations to the CSF1R signalling network, we used netAnalysis_signallingRole_network().

## Cell culture
### Primary microglial culture
We isolated microglia by immunopanning and cultured in serum-free conditions as per our previous protocol (Zia et al., 2022). Briefly, we dissected cortices from P5-P7 CD1 mice (Charles River Laboratories, strain 022) and homogenized tissue with gentleMACS Neural Tissue Dissociation Kit P (Miltenyi Biotec, 130-092-628). We inhibited residual enzymatic activity by pelleting the homogenates through a low-to-high concentration ovomucoid gradient whereby the cell suspension present in the lower density ovomucoid layer, composed of ovomucoid (Worthington Biochemical, LS003086), bovine serum albumin (BSA; Sigma-Aldrich, A8806), DNase (8.3 µg/ml; Worthington Biochemical, LS002007) and DPBS with calcium and magnesium (Gibco, 14040182) was passed through a more dense, high ovomucoid layer of the same composition, though with twice as much ovomucoid and BSA and a higher concentration of DNase (20.8 µg/ml). We next resuspended cells in a solution of 0.02% BSA (Sigma-Aldrich, A4161), DNase (33 µg/ml; Worthington Biochemical, LS002007), DPBS with calcium and magnesium (Gibco, 14040182), and filtered in series through 70 µm and 40 µm filters. We passed the homogenate over an anti-CD11b (clone M1/70; Invitrogen, 14-0112-85)-coated dish to capture microglia. We trypsinized (30,000 units/ml; Sigma-Aldrich, T9935) CD11b-bound cells for 4-6 min at 37°C and dislodged cells via repeated pipetting and plated 10,000 microglia/well in serum-free microglial growth media on poly-D-lysine (10 mg/ml; Sigma-Aldrich, P6407)-coated 96-well culture plates (Falcon, 353219). Microglial growth media consisted of a Neurobasal Media base (Gibco, 21103049), Gem 21 supplement (1×; GeminiBio, 400-160-010), N2 supplement (1×; Thermo Fisher Scientific, 17502-048), lipated BSA (1 mg/ml; Thermo Fisher Scientific, 11020021), Glutamax (2 mM; Gibco, 35050-061), sodium pyruvate (1 mM; Gibco, 11360-070), sodium chloride (50 mM; Sigma-Aldrich, S5150), D-biotin (10 ng/ml; Sigma-Aldrich, B4639) and lactic acid (0.2%; Sigma-Aldrich, L1250). To ensure serum-free microglial survival, we supplemented media with CSF1 (10 ng/ml; Gibco, 315-02), transforming growth factor beta 2 (2 ng/ml; Gibco, 100-35B), cholesterol (1.5 µg/ml; Avanti Polar Lipids, 700000P), heparan sulphate (1 µg/ml; Galen Laboratory Supplies, GAG-HS01), oleic acid (100 ng/ml; Cayman Chemicals, 90260) and gondoic acid (1 ng/ml; Cayman Chemicals, 20606), as previously described (Bohlen et al., 2017). Microglia were kept at 37°C, 10% $CO_2$ for 5 days before treatments.

### Microglial viability and proliferation
To assess microglial viability, we treated microglia with a cocktail of NucBlue™ Live ReadyProbes™ Reagent (Hoechst 33342; Invitrogen, R37605), CalceinAM (2 µg/ml; Invitrogen, C3099) and propidium iodide (4 µg/ml; Invitrogen, P3566) diluted in Neurobasal Media (Gibco, 21103049). After 10 min at 37°C, we imaged microglia on an ImageXpress Micro (Molecular Devices; University of Alberta Cell Imaging Facility) at 20× magnification. We measured fluorescent cell scoring in MetaXpress (Molecular Devices) to quantify CalceinAM and propidium iodide fluorescence as live and dead, respectively.

To determine microglial purity, we fixed microglia after 8 days *in vitro* with 4% paraformaldehyde for 10 min at room temperature. We washed microglia with DPBS and blocked with 10% normal donkey serum (Millipore-Sigma, D9663) diluted in 0.1% Triton X-100 (Fisher Scientific, BP151-500) and DPBS with calcium and magnesium (Gibco, 14040-133) for 45 min at room

temperature. Without washing, we added the primary antibodies rat anti-CD11b (1:100; Invitrogen, 14-0112-85) overnight at 4°C. We washed off primary antibodies with DPBS and added secondary antibodies for 2 h at room temperature before a final wash with DPBS. We assessed the percentage of total CD11b nuclei relative to the total nuclei as a measure of microglial purity on an ImageXpress Pico and conducted multiwavelength cell scoring analyses in CellReporterXpress (Molecular Devices).

For dose response experiments, we created tenfold dilutions of CSF1 (Gibco, 315-02) and IL34 (R&D Systems, 5195-ML) in microglial growth media. We excluded CSF1 from our growth media to ensure that CSF1R was available for binding by CSF1 or IL34. We similarly prepared treatments for predicted CellChat factors ANGPTL4 (R&D Systems, 4880-AN), CCL3 (Gibco, 250-09), CCL4 (Gibco, 250-32), CX3CL1 (Gibco, 300-31), GAL9 (R&D Systems, 3535-GA), GAS6 (R&D Systems, 986-GS-025/CF), MDK (Gibco, 315-25), PTN (R&D Systems, 6580-PL) and TGFβ2 (Gibco, 100-35B) in microglial base media that lacked CSF1 and TGFβ2 at 10 ng/ml. We additionally prepared tenfold dose responses of each in microglial growth media containing CSF1 and TGFβ2. We treated microglia with the dilution series in triplicate for 72 h. After 72 h, we fixed microglia with 4% paraformaldehyde, blocked with 10% normal donkey serum diluted in 0.1% Triton X-100 (Fisher Scientific, BP151-500) and DPBS with calcium and magnesium (Gibco, 14040-133) for 45 min at room temperature. We added rabbit anti-CD11b (1:200; Novus Biologicals, NB110-89474) or rat anti-CD11b (1:100; Invitrogen, 14-0112-85), and mouse anti-KI67 (1:200; clone B56, BD Pharmingen, 550609), diluted in DPBS and incubated at 4°C overnight. We washed microglia the following day with DPBS (Gibco, 14040-133) and added secondary antibody solution with combinations of anti-rabbit Alexa Fluor 488 (1:400; Jackson ImmunoResearch, 711-546-152) or anti-rat Alexa Fluor 488 (1:400; Jackson ImmunoResearch, 712-546-153), and anti-mouse Alexa Fluor 647 (1:400; Jackson ImmunoResearch, 715-606-151) conjugated $F(ab)_2$ fragment secondary antibodies and DAPI (5 µg/ml; Invitrogen, D1306) for 2 h at room temperature. We washed microglia with DPBS (Gibco, 14040-133) and imaged cells on an ImageXpress Pico (Molecular Devices) or ImageXpress Micro (Molecular Devices; University of Alberta Cell Imaging Facility). All analyses were conducted in CellReporterXpress or MetaXpress using the multiwavelength cell scoring analysis tool.

## Image analysis and density maps
We counted microglia for all microglial proliferation analyses in either QuPath (Bankhead et al., 2017), Fiji (Schindelin et al., 2012) or ZenLite (Zeiss). We quantified all RNAscope datasets in QuPath by training a neural network to conduct an unbiased assessment of transcript expression within nuclei. Thresholds for *Csf1*- or *Il34*-positive nuclei were determined based on the fluorescence intensity corresponding to five transcripts within nuclei. For the developmental timeline of microglial proliferation, we counted and averaged a total of three to four images – two from each hemisphere – from within CA1 and from all layers of the somatosensory cortex. For RNAscope analyses, we averaged counts in regions of interest of CA1 and all layers of the somatosensory cortex from each hemisphere. For the timeline of developmental microglial proliferation in IL1 knockouts, we averaged counts of microglia and proliferative microglia throughout all regions of the hippocampus and all layers of the somatosensory cortex across regions of interest in one or two brain sections. Each assessor was unaware of animal age or genotype prior to analysis. We conducted all cell culture analyses in CellReporterXpress (Molecular Devices) using the multiwavelength cell scoring tool.

We created density maps of *Csf1* and *Il34* transcripts in QuPath. We used the cell detection tool to positively identify each fluorescent mRNA transcript across the brain scan followed by the density maps tool with a Gaussian-weighted density type and a density radius of 50 pixels. We exported density maps to Fiji, cropped and scaled images to the same size, created an image stack, and created a median intensity *z*-project for each.

## Statistical analyses
We plotted all graphs for histology and cell culture experiments with, and conducted all statistical testing in, GraphPad Prism 10. Data are presented as mean±s.e.m. with multiple group comparison significance assessed with

one-way or two-way ANOVAs as appropriate with a Tukey or Dunnett's post-hoc test for significance (*$P<0.05$, **$P<0.01$, ***$P<0.001$, ****$P<0.0001$).

## Acknowledgements
We acknowledge and thank the Cell Imaging Core and the Cell Imaging Facility of the Faculty of Medicine and Density, University of Alberta for their aid, expertise and access to microscopes and the University of Alberta Health Sciences Laboratory Animal Services for their housing and care of animals used in these studies. Schematics were made in BioRender.

## Competing interests
The authors declare no competing or financial interests.

## Author contributions
Conceptualization: B.P.H., B.J.K., J.R.P.; Data curation: B.P.H., S.Z., E.H., T.L., S.F., R.M., J.R.P.; Formal analysis: B.P.H., S.Z., E.H., M.K., T.L., S.F., R.M., K.V.L.; Funding acquisition: B.J.K., J.R.P.; Investigation: B.P.H., S.Z., E.H., T.L., S.F., R.M., K.V.L., A.C.-M., F.B., S.L., J.R.P.; Methodology: B.P.H., K.V.L., J.R.P.; Project administration: B.P.H., S.Z., E.H., T.L., S.F., R.M., K.V.L., A.C.-M., F.B., B.J.K., S.L., J.R.P.; Resources: B.P.H., S.L., J.R.P.; Supervision: B.J.K., M.S.C., S.L., J.R.P.; Writing – original draft: B.P.H., J.R.P.; Writing – review & editing: B.P.H., B.J.K., S.L., J.R.P.

## Funding
B.P.H. was funded by a Natural Sciences and Engineering Research Council of Canada (NSERC) Masters and Doctoral scholarship from the University of Alberta. S.Z. was funded by an NSERC Masters scholarship and scholarships from the University of Alberta. R.M. was funded by an NSERC undergraduate student research award. J.R.P. was supported by funds from Canada Research Chairs (Tier 2) in Glial Neuroimmunology. This study was funded by operating grants from NSERC (RGPIN-2019-04533). Open Access funding provided by the University of Alberta. Deposited in PMC for immediate release.

## Data and resource availability
All relevant data and details of resources can be found within the article and its supplementary information.

## Peer review history
The peer review history is available online at https://journals.biologists.com/dev/lookup/doi/10.1242/dev.204610.reviewer-comments.pdf

## Special Issue
This article is part of the Special Issue 'Lifelong Development: the Maintenance, Regeneration and Plasticity of Tissues', edited by Meritxell Huch and Mansi Srivastava. See related articles at https://journals.biologists.com/dev/issue/152/20.

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
