## [Peer Review File · Development (Cambridge, England)]

CSF1R ligands promote microglial proliferation but are not the sole regulators of developmental microglial proliferation

Brady P. Hammond, Sameera Zia, Eugene Hahn, Margarita Kapustina, Tristan Lange, Sarah Friesen, Rupali Manek, Kelly V. Lee, Adrian Castellanos-Molina, Floriane Bretheau, Mark S. Cembrowski, Bradley J. Kerr, Steve Lacroix and Jason Plemel

DOI: 10.1242/dev.204610

Editor: Francois Guillemot

Review timeline

Original submission: 23 December 2024

Editorial decision: 3 February 2025

First revision received: 24 April 2025

Accepted: 7 May 2025

Original submission

First decision letter

MS ID#: dev.204610

MS Title: CSF-1R ligands promote microglial proliferation but are not the sole regulators of developmental microglial proliferation

Authors: Brady P. Hammond; Sameera Zia; Eugene Hahn; Tristan Lange; Sarah Friesen; Rupali Manek; Kelly V. Lee; Adrian Castellanos-Molina; Floriane Bretheau; Bradley J. Kerr; Steve Lacroix; Jason Plemel

Article Type: Research Article

Dear Dr Plemel,

I have now received all the referees' reports on the above manuscript, and have reached a decision. The referees' comments are appended below, or you can access them online: please go to:

As you will see, the referees express great interest in your work, but have some significant criticisms and recommend a substantial revision of your manuscript before we can consider publication. In particular, they request that you improve the presentation of the data (including providing information on the regions and layers where microglia quantification has been performed), that you increase the sample size for the RNAscope data, that you use a more specific microglia marker than Iba1, etc. If you are able to revise the manuscript along the lines suggested, which may involve further experiments, I will be happy to receive a revised version of the manuscript. Your revised paper will be re-reviewed by one or more of the original referees, and acceptance of your manuscript will depend on your addressing satisfactorily the reviewers' major concerns. Please also note that Development will normally permit only one round of major revision. If it would be helpful, you are welcome to contact us to discuss your revision in greater detail. Please send us a point-by-point response indicating your plans for addressing the referees' comments, and we will look over this and provide further guidance.

Please attend to all of the reviewers' comments and ensure that you clearly highlight all changes made in the revised manuscript. Please avoid using 'Tracked changes' in Word files as these are lost in PDF conversion. I should be grateful if you would also provide a point-by-point response detailing

how you have dealt with the points raised by the reviewers in the 'Response to Reviewers' box. If you do not agree with any of their criticisms or suggestions please explain clearly why this is so.

Reviewer 1: The authors explore murine microglial proliferation during development and investigate potential molecular factors that could regulate it. They begin by characterizing microglial proliferation from birth to P28, focusing on two key regions: the cortex and the hippocampus. Through a bioinformatics approach leveraging a published transcriptomic dataset at the end of the first postnatal week while microglia are highly proliferative, they identify receptor-ligand pairs that may regulate microglial proliferation. They subsequently test a dozen ligands on primary microglial cultures revealing that only CSF1R ligands, CSF1 and IL34, impact microglial proliferation.

The authors examine the expression of CSF1 and IL34 using ELISA and RNAscope in the CA1 region of the hippocampus and the somatosensory cortex along development. They report a peak in the expression of both ligands at P10, with CSF1 being expressed at much lower levels than IL34. By analyzing the expression of these ligands in various neural populations using the same published transcriptomic dataset, they identify fibroblasts, pericytes, and Cajal cells as major sources of IL34 at P7.

Finally, they investigate microglial density in full knockout mice for Il1a and Il1b. They observe reduced microglial density during development, which is associated with increased proliferation but without changes in CSF1 or IL34 expression.

This study presents interesting findings but lacks some information, in particular some aspects of the methodology and presentation should be improved for greater clarity and coherence.

Microglial density in the cortex and hippocampus is known to be highly heterogeneous across layers and subregions during development. The authors should clearly specify the exact areas where they quantified microglial density and proliferation, both at the regional level (e.g., cortex, CA1, dentate gyrus) and at the layer level, especially in the cortex where microglial densities are highly heterogeneous.

The RNAscope analysis for *Csf1* seems to rely on an $n=1$, which is insufficient to support robust conclusions. Increasing the sample size is essential to strengthen the validity of the findings.

The section on the IL1 knockouts is intriguing but feels disconnected from the rest of the manuscript and the interpretation of the results is unclear. The findings lack a clear narrative linking them to the earlier parts of the study. Additionally, the authors should clearly differentiate what is novel from what is already established in the field to better contextualize the significance of their results.

Reviewer 2

SUMMARY OF THE ADVANCE MADE IN THIS PAPER AND ITS POTENTIAL SIGNIFICANCE TO THE FIELD

The manuscript "CSF-1R ligands promote microglial proliferation but are not the sole regulators of developmental microglial proliferation" by Hammond et al. is posed to identify ligand-receptor pairs that regulate microglial proliferation across development using a published scRNA database and a bioinformatic approach, CellChat, an interesting approach whose limitations should be more openly discussed. Next the authors chose to validate the candidates in culture, again an approach with limitations that must be acknowledged. Overall, the results are interesting, most of the conclusions are supported by the data, and the paper opens interesting questions about yet-to-be-discovered mechanisms controlling microglial developmental proliferation.

SUGGESTIONS TO AUTHORS

Major issues

1, The CellChat approach is very interesting, but to assess the reliability of the results, the scRNASeq dataset utilized must be more clearly described: ages, conditions, number of cells, depth of sequencing, etc. In addition, since there are other similar developmental databases available, the authors should justify why only this one was analyzed (some expression data, but not the CellChat data, is validated in the Hammond database in Figure 5). The number of microglia is particularly relevant to address the robustness of the results. A final point to clarify is whether the

authors filtered the ligand-receptor dataset for released factors, since no membrane tethered signals were found.

2, The link between the effects of CSF1 and IL34 on microglial proliferation and survival is interesting but not fully resolved. The sentence "Given that 1 ng/mL CSF-1 did not enhance microglial proliferation, but did sustain viability, CSF-1 may variably support microglial viability and proliferation at different levels" (line 205) is unclear. Is it possible that only proliferative cells survive, resulting in an observed increase in proliferation, without directly boosting proliferation? Although this point is not easy to resolve experimentally, the authors need to speculate on this issue more clearly.

3, The decision to study the role of the selected factors in vitro should be justified. Specifically, the authors need to state the limitations of such method, for it may explain why many ligand-receptor pairs identified by CellChat do not seem to control proliferation. In addition, the authors should discuss that those pairs may regulate other microglial functions apart from proliferation.

4, This reviewer disagrees with the conclusion that "Peak CSF-1 and IL-34 protein and transcript levels align with microglial proliferation" (line 210). It is unfortunate that the analysis of proliferation (Figure 1) was not performed exactly at the same time points that the analysis of CSF protein and mRNA (Figure 4), preventing linear regression analyses. However, the main point is that proliferation drops after P7 but both proteins peak later. The authors need to discuss this discrepancy more clearly.

5, To compare the RNAscope distribution of Il34 and Csf1 (Figure 4) with the expected producing cell, based on their CellChat analysis (Figure 2), the authors show the expression of these two transcripts on the UMAP of the different clusters. However, there seems to be a major difference with the RNAscope data: CellChat identifies IL34 in fibroblasts, pericytes and Cajal Retzius cells, but the RNAscope shows labeling in what seems to be the granule cell layer (Figure 4D). This lack of correlation, if confirmed, could suggest the lack of robustness of the CellChat analysis or a problem with the Joglekar database. Another potential issue that should be acknowledged is that the CellChat analysis is based on a single time point (P7), and thus interactions of microglia and other cells over time may be missed.

6, The authors need to justify the selection of the IL1a and IL1b KO mice as a model to study aberrant microglial proliferation, because: 1, this signaling pathway was not identified in their CellChat analysis; 2, one would expect this pathway to be involved in inflammation-induced proliferation but not basal proliferation; and 3, to be best of my knowledge, the IL1 and IL34 pathways are unrelated. Thus, there is no rationale to assume that IL1-mediated proliferation should be mediated by IL34, nor that IL34 levels would explain proliferation in IL1 KO mice.

7, The statement that "Cx3cl1, rather than directly inducing microglial proliferation, instead recruits microglia to, and/or facilitates transit within, the developing CNS and thus situates microglia within the proliferative niche of the developing CNS" (line 333) is beyond any data provided and excessively speculative. This reviewer is puzzled by the concept of "proliferative niche of the developing CNS" - microglia proliferate throughout the parenchyma, not in proliferative niches. The sentence should be

Minor issues:

1. Whether microglia "actively phagocytose synapses" (line 65) has been recently questioned in Eyo and Molofsky Science 2023 (PMID: 37708287) and Pereira-Iglesias et al., Nat Neurosci 2024 (PMID: 39663381), and should be rephrased.
2. The sentence on "microglia (...) precipitate depressive-like and anxiety-like behaviors in adulthood (Delpuch et al., 2016)" does not reflect the results from said paper, which is largely correlational. The sentence should be rewritten (line 73).
3. There seems to be a grammatical error in the sentence "(...) microglial proliferation in development results from a mitogen directly promotes microglial proliferation" (line 110)
4. There is a typographical error in the sentence "a publicly available scRNAseq datasets" (line 144).
5. Figure 3: show the Ki67 channel separated, in addition to the merged image.

- 6, The sentence "that CSF-1 promotes equivalent levels of proliferation with or without TGF- β 2, CSF-1 likely promotes proliferation independently from survival." is confusing. What experimental groups do the authors compare to test the effect of CSF "with or without TFGb2"?
- 7, Figure legends should state the meaning of asterisks and the statistic test used in each comparison.
- 8, The color code of the UMAP clusters is not explained in the figure legend (Figure 2 and 5).

Reviewer 3

SUMMARY OF THE ADVANCE MADE IN THIS PAPER AND ITS POTENTIAL SIGNIFICANCE TO THE FIELD

In the study the authors investigated the factors driving microglia proliferation across key developmental stages. Using immunofluorescence, they characterized microglia density and proliferation in the cortex and hippocampus from P0 to P30. To identify potential drivers of proliferation, the authors analyzed a previously published transcriptomic dataset, pinpointing 12 candidate ligand-receptor pairs. In vitro assays revealed that two ligands appreciably enhanced microglia proliferation (CSF1 and IL-34). These ligands were subsequently evaluated in whole brain homogenates using ELISA and in tissue slices via immunofluorescence and RNAscope to assess spatiotemporal expression in the cortex and hippocampus. These experiments identified IL-34 as the only ligand with consistent and significant expression across all developmental time points in each assay. However, the authors propose that other identified ligands may act synergistically or indirectly contribute to creating a permissive environment for microglia proliferation.

This study adds modestly to prior literature on the regulation of microglia development and proliferation by CSF1R signaling. While seminal work has identified that CSF1 and IL-34 contribute to microglia numbers and survival during development, here the authors question if these ligands also induce proliferation. As the authors' work makes clear, it is challenging to distinguish ligand effects on proliferation when there is also a requirement for survival. Also, there are concerns about the author's interpretation about the source of ligands. Multiple prior studies show that IL-34 is highly expressed by neurons in the developing brain, but that is not evident from the CellChat analysis using the published transcriptomic dataset, raising concerns about the dataset used.

SUGGESTIONS TO AUTHORS

There are some areas in the methods, figures, and experimental design that require clarification. Addressing these concerns will improve the manuscript's clarity and reproducibility.
Major Comments:

- 1) **Microglia Marker Limitations (Figure 1):** Iba1 expression is not exclusive to microglia, making it difficult to definitively attribute proliferation or increased density to microglia alone. The observed proliferation might involve myeloid cells recruited from outside the microglia population, raising questions about whether the expansion is solely due to resident microglia or includes recruited macrophages. This issue should be addressed by using a more specific microglia marker such as Sall1 or P2ry12.
- 2) **Significance of Cell Proliferation (Figure 3):** The authors identify CSF1, IL-34, and ANGPTL4 as ligands that promote proliferation in vitro. However, CellChat predicts minimal "communication probability" for CSF1 and ANGPTL4. This discrepancy between in vitro results and computational predictions should be addressed in the discussion, particularly in light of the high communication probability of Ptn, but limited impact on proliferation thus no further study. As noted above, the cell type enrichment from this analysis does not align with prior published work showing high enrichment of IL-34 in neurons.
- 3) **Methodology and Media Use (Figure 3):** Lines 182-186 could be rewritten as the methodology for switching out different medias and timing is confusing. In addition, the use of MGM media for only a subset of the mitogens identified raises some concerns:
 - * Since TGFB2 and CSF1 are already included in the serum-free media, this potentially confounds the interpretation of additional mitogen effects. Ie---the impact of other mitogens

added may not be appreciated in the presence of CSF1 in the serum-free media, since it was shown to be sufficient to increase microglia proliferation in figure 3C. Why not compare all mitogens across the same media conditions? If TGFB2 and CSF1 are deemed essential for microglial survival, how do microglia survive in the base media, which lacks these factors?

* Would it be possible to clarify the significance of comparing MGM in the 3C figure? Was this media used in any of the assays?

* Please clarify the concentrations of TGFB2 and CSF1 in the serum-free condition versus the additional amounts included in the experimental conditions. While Figure 3D specifies the concentration of the added mitogen, this information is absent for Figure 3C, making comparisons difficult.

* The axes in Figures 3C and 3D are not consistent. Standardizing the axes would facilitate easier comparison between the data.

* Why is CD11b used as the marker for identifying "microglia" in this figure, while IBA1 is used in other figures? CD11b is less specific to microglia compared to IBA1 and may also label other myeloid cells.

* In Figure 3D, CSF1, MGM, and IL34 are not listed, despite their relevance and further exploration of impact on microglial proliferation. Could the authors clarify why these ligands/MGM were omitted from this graph? Additionally, there is no rationale provided for the specific selection of ligands for the dose-response curves in Figures 3E and 3F. Including a brief explanation for the choice of ligands would be helpful. Ie—why was ANGPTL4 not examined even though it demonstrates the highest fold change out of all the ligands?

4) Confounding Results (Figure 3): The claim that IL-34 increases proliferation should be reconsidered, as the data for this is not clearly presented (IL-34 is not depicted in figure 3C or 3D compared to other ligands). The overlap of CSF1 present in the serum-free media could confound results of the addition of IL34 since they both act on the same receptor. The authors should discuss how this may influence the interpretation of the role of IL34 on proliferation.

5) Inflammation Model (Figure 6): The rationale for using IL-1B and IL-1a KO mice to model aberrant inflammation seems insufficient, especially since the authors cite literature that microglia may not express the receptors for these cytokines, suggesting indirect effects. There is a potential inconsistency between the interpretation of proliferation and density in the IL-1B KO model. Increased proliferation with decreased density contradicts earlier claims from Figure 1 that attributed increased density to proliferation. Further clarification is needed regarding how these findings align.

6) Strain Discrepancies: The authors use different mouse strains for ELISA assays at different time points (P0, P7, P30). It would be helpful to clarify why this approach was chosen and how strain-specific differences might influence results. Furthermore, the use of the iDTR mouse line is mentioned in methods, but its specific role and representation in the data/main text are not clearly outlined. Why are both B6 and iDTR mice used and at different timepoints?

7) In Vitro vs In Vivo Mouse Strain Differences: The authors use cells from CD1 mice for in vitro experiments, while in vivo experiments utilize B6/iDTR mice. It would be valuable to explain why these strains are used differently and whether strain variability could impact results and interpretation across in silico, in vitro, and in vivo assays.

Minor Comments:

8) Figure 1 (Y-Axis Labeling): The Y-axis labels for panels C and F are unclear. Clarifying the measurement scale and confirming whether 0.2 microglia per mm² is correct would help prevent misinterpretation.

9) Significance Markers: It may be helpful to report exact p-values rather than using stars for significance in the graphs to improve clarity and rigor.

10) Methodological Clarification (Figure 1): The figure legend should specify the number of sections per brain examined, which is currently missing in the methods section.

11) Isolated Cell Identity (Figure 3): How are isolated microglia being confirmed? Is CD11b the sole marker used? CD11b is not microglia-specific and can be expressed by other myeloid cells, such

as macrophages. Incorporating additional markers like P2RY12 or Sall1 could strengthen the identification of microglia. Additionally, the morphology of the cells shown in the image does not align with the characteristic ramified structure typically observed in microglia under homeostatic conditions.

12) In Situ vs Protein (Figure 5): There seems to be a discrepancy between in situ staining and protein quantification, for the time points used. More clarity on this would be beneficial.

13) IL-34 Staining (Figure 5): It would be useful to define the threshold for positive IL-34 staining and provide a more detailed explanation of the staining pattern in the figure legend. It is listed in the methods, but it would be helpful to have in the results or figure legend description as well.

14) Image Analysis (Figure 5): The methods should specify how many images per mouse were analyzed/averaged and how regions of interest in the cortex and hippocampus were selected. Providing these details would improve reproducibility.

First revision

Author response to reviewers' comments

Reviewer 1:

The authors explore murine microglial proliferation during development and investigate potential molecular factors that could regulate it. They begin by characterizing microglial proliferation from birth to P28, focusing on two key regions: the cortex and the hippocampus. Through a bioinformatics approach leveraging a published transcriptomic dataset at the end of the first postnatal week while microglia are highly proliferative, they identify receptor-ligand pairs that may regulate microglial proliferation. They subsequently test a dozen ligands on primary microglial cultures revealing that only CSF1R ligands, CSF1 and IL34, impact microglial proliferation.

The authors examine the expression of CSF1 and IL34 using ELISA and RNAscope in the CA1 region of the hippocampus and the somatosensory cortex along development. They report a peak in the expression of both ligands at P10, with CSF1 being expressed at much lower levels than IL34. By analyzing the expression of these ligands in various neural populations using the same published transcriptomic dataset, they identify fibroblasts, pericytes, and Cajal cells as major sources of IL34 at P7.

Finally, they investigate microglial density in full knockout mice for Il1a and Il1b. They observe reduced microglial density during development, which is associated with increased proliferation but without changes in CSF1 or IL34 expression.

Comment:

This study presents interesting findings but lacks some information, in particular some aspects of the methodology and presentation should be improved for greater clarity and coherence.

1. Microglial density in the cortex and hippocampus is known to be highly heterogeneous across layers and subregions during development.

Author: Thank you for noting this. We have added the following into our results and methods to account for these discrepancies.

Lines 134-136:

In the developing somatosensory cortex and CA1 of the hippocampus, microglia proliferated robustly during the first two postnatal weeks, with a cessation of nearly all microglial proliferation by P14 (Fig. 1A-F).

2. The authors should clearly specify the exact areas where they quantified microglial

density and proliferation, both at the regional level (e.g., cortex, CA1, dentate gyrus) and at the layer level, especially in the cortex where microglial densities are highly heterogeneous.

Author: Thank you for this request. We have adjusted the Materials and Methods to specify this as follows:

Lines 749-756

For the developmental timeline of microglial proliferation, we counted and averaged a total of three to four images—two from each hemisphere—from within CA1 and from all layers of the somatosensory cortex. For RNAscope analyses, we averaged counts in ROIs of CA1 and layers all layers of the somatosensory cortex from each hemisphere. For the timeline of developmental microglial proliferation in IL-1 knockouts, we averaged counts of microglia and proliferative microglia throughout all regions of the hippocampus and all layers of the somatosensory cortex across ROIs in 1-2 brain sections.

3. The RNAscope analysis for *Csf1* seems to rely on an $n=1$, which is insufficient to support robust conclusions. Increasing the sample size is essential to strengthen the validity of the findings.

Author: Thank you for noting this. We have redone these analyses with a greater sample size (see Fig. 4L).

4. The section on the IL1 knockouts is intriguing but feels disconnected from the rest of the manuscript and the interpretation of the results is unclear. The findings lack a clear narrative linking them to the earlier parts of the study.

Author: Thank you for this comment. We have adjusted the text as follows to provide a clearer rationale for using IL1 knockouts as models of abnormal developmental proliferation and to better contextualize our interpretations of results.

Lines 344-357:

Considering the overlap of microglial proliferation with the increasing expression of *Csf1* and *Il34* throughout development, and the ability of both factors to boost proliferation in culture, we next aimed to explore whether the expression of these mitogens is altered alongside aberrant developmental microglial proliferation. CSF-1R “loss-of-function” models have developmental abnormalities and lack microglia (B. P. Hammond, Manek, Kerr, Macauley, & Plemel, 2021). CSF-1 and IL-34 are critical for microglia survival and, therefore, challenging to disentangle their proliferative and survival-promoting roles in vivo. Fortunately, microglial proliferation has been more widely explored in models of adulthood repopulation following microglial depletion, wherein the depletion of microglia initiates a robust, CNS-wide proliferative phase, analogous to development, that permits microglia to populate the entire CNS (Bruttger et al., 2015; Elmore et al., 2014; Huang et al., 2018). Notably, interleukin-1 receptor-1 signalling partially regulates this proliferation following microglial depletion (Bruttger et al., 2015). Given the similarity of microglial proliferation between repopulation models and development, we reasoned that interleukin-1 signalling may similarly regulate developmental microglial proliferation.

5. Additionally, the authors should clearly differentiate what is novel from what is already established in the field to better contextualize the significance of their results.

Author: Thank you for this suggestion. We have adjusted the text of our discussion as follows to directly contextualize the novelty of our study.

Lines 381-400:

The developmental proliferation of microglia is critical for the appropriate development of the CNS. Here, we histologically demonstrate that murine microglia proliferate for the first two postnatal weeks, with a peak at P7, to establish a homeostatic density that is maintained into adulthood, thus expanding upon previous assessments of developmental microglial proliferation via flow cytometry (Ginhoux et al., 2010; Nikodemova et al., 2015) and histological assessments of microglial densities (Kim et al., 2015). In testing factors predicted to signal onto microglia at the

peak of developmental proliferation, we found that the CSF-1R ligands—CSF-1 and IL-34—boost proliferation in primary, serum-free microglia. While CSF-1R ligands induced proliferation has been described previously, to the best of our knowledge, we are the first to disentangle the role of CSF-1R ligands in survival and proliferation in serum-free conditions that generate microglia similar to *in vivo*, developmental conditions (Bohlen et al., 2017). Many of the remaining factors that we identified in our receptor-ligand analysis of developing microglia have no known impact on microglia, which raises intriguing questions regarding their potential developmental functions. We also found that both CSF-1 and IL-34 increase throughout development in distinct spatial and temporal patterns as has been recently described (Devlin et al., 2024), and that IL-1 α or IL-1 β knockout mice have abnormal developmental microglial proliferation, which was not associated with compensatory alterations in *Csf1* and *Il34* transcript expression. These findings support the idea that, while microglia are capable of robust proliferation at distinct developmental timepoints and in response to highly specific mitogenic signalling, at least one additional regulatory factor of developmental proliferation remains to be discovered.

Reviewer 2: SUMMARY OF THE ADVANCE MADE IN THIS PAPER AND ITS POTENTIAL SIGNIFICANCE TO THE FIELD

The manuscript "CSF-1R ligands promote microglial proliferation but are not the sole regulators of developmental microglial proliferation" by Hammond et al. is posed to identify ligand- receptor pairs that regulate microglial proliferation across development using a published scRNA database and a bioinformatic approach, CellChat, an interesting approach whose limitations should be more openly discussed. Next the authors chose to validate the candidates in culture, again an approach with limitations that must be acknowledged. Overall, the results are interesting, most of the conclusions are supported by the data, and the paper opens interesting questions about yet-to-be-discovered mechanisms controlling microglial developmental proliferation.

SUGGESTIONS TO AUTHORS

Major issues

1, The CellChat approach is very interesting, but to assess the reliability of the results, the scRNASeq dataset utilized must be more clearly described: ages, conditions, number of cells, depth of sequencing, etc. In addition, since there are other similar developmental databases available, the authors should justify why only this one was analyzed (some expression data, but not the CellChat data, is validated in the Hammond database in Figure 5). The number of microglia is particularly relevant to address the robustness of the results. A final point to clarify is whether the authors filtered the ligand-receptor dataset for released factors, since no membrane tethered signals were found.

Author: Thank you for this request. We have explicitly outlined our criteria for selecting this dataset in the text. While many other developmental scRNAseq databases do exist, all failed at least one of our criteria (eg. age or restricted to specific cell lineages). The request for clarification on released factors is an important one. Given that we were working to identify a mitogen released by another cell type to induce microglial proliferation, we filtered the CellChat database to limit results to "Secreted Signalling" factors.

Lines 162-173:

We searched the Gene Expression Omnibus (GEO) database for a single-cell RNA sequencing dataset containing as many cell types as possible in the developing brain when microglia are highly proliferative, ideally between P7 and P10. As a secondary criterion, we aimed to find a dataset where cells were separated based on brain region. With these criteria, we identified the dataset published by Joglekar and colleagues (Joglekar et al., 2021). They prepared a single-cell RNA sequencing dataset from the prefrontal cortices and hippocampi of P7 mice. We first conducted quality control to exclude doublets or dying cells via analysis of total gene and mitochondrial gene counts, respectively (Fig. S1B) before pooling the two cortical replicates (n=3 cortices total) each of the two cortical datasets and each of the two hippocampal datasets (n=3 hippocampi total) into a single cortical and a single hippocampal dataset. With this, we had 6400 cells in the cortical dataset, of which 409 were microglia, and 14077 cells in the hippocampal dataset, of which 483

were microglia.

Lines 182-184:

We performed CellChat to explore potential signalling onto microglia from other cell populations/subsets and filtered the predicted results to “Secreted Signalling” factors of the CellChat database.

2, The link between the effects of CSF1 and IL34 on microglial proliferation and survival is interesting but not fully resolved. The sentence “Given that 1 ng/mL CSF-1 did not enhance microglial proliferation, but did sustain viability, CSF-1 may variably support microglial viability and proliferation at different levels” (line 205) is unclear. Is it possible that only proliferative cells survive, resulting in an observed increase in proliferation, without directly boosting proliferation? Although this point is not easy to resolve experimentally, the authors need to speculate on this issue more clearly.

Author: Thank you for raising this intriguing point. We have adjusted the text as follows to include this possibility and to more effectively convey our interpretations of results.

Lines 252-260:

Given that 1 ng/ml CSF-1 was sufficient to sustain microglial viability (Fig 3E-F), but a concentration of 10 ng/ml of CSF-1 was required to initiate appreciable microglial proliferation (Fig 3D), CSF-1 may regulate distinct aspects of microglial biology at differing concentrations. Alternatively, CSF-1 ligands may require a higher concentration to enhance the survival of proliferative cells.

3, The decision to study the role of the selected factors *in vitro* should be justified. Specifically, the authors need to state the limitations of such method, for it may explain why many ligand-receptor pairs identified by CellChat do not seem to control proliferation. In addition, the authors should discuss that those pairs may regulate other microglial functions apart from proliferation.

Author: Thank you for this request. We have elaborated on our decision to explore these factors in culture in the text.

Lines 201-215:

To explore whether the ligand-receptor pairings predicted to signal onto P7 microglia may boost microglial proliferation, we elected to test these factors on serum-free, primary microglia isolated from CD1 mice. Although primary CD1 microglia display subtly elevated inflammatory responses in response to LPS stimulation relative to C57BL/6 counterparts both *in vitro* and *in vivo* (Nikodemova & Watters, 2011) the greater litter sizes, and thus greater yield of microglia, made them a suitable model for our cell culture experiments. Further, the use of outbred mouse stocks for studying immunological mechanisms has been suggested to be a critical “next step” in translating findings from the bench to bedside (Enriquez, Mims, Trasti, Furr, & Grisham, 2020). While cell culture is limited in its ability to recapitulate the suite of interactions that occur *in vivo*, it provides a valuable, high-throughput tool to assay several potential mechanisms. Additionally, serum-free microglia more faithfully resemble *in vivo* microglia than previous culture methods (Bohlen et al., 2017). We isolated primary mouse microglia from the P5-P7 cortex by immunopanning (Fig. 3A), which yielded highly pure (on average 95.4%) CD11b⁺ microglial cultures, of which nearly 100% were also Tmem119⁺, by day 7-8 *in vitro* (Fig. S3A- C).

Lines: 256-260:

While most predicted factors failed to induce microglial proliferation in serum-free culture, these factors may regulate other aspects of developmental microglial biology, not proliferation. Alternatively, cultured microglia may be rendered incompetent to respond to these factors by downregulating receptors that would otherwise be expressed within the CNS.

4, This reviewer disagrees with the conclusion that “Peak CSF-1 and IL-34 protein and transcript levels align with microglial proliferation” (line 210). It is unfortunate that the analysis of proliferation (Figure 1) was not performed exactly at the same time points that the analysis of CSF protein and mRNA (Figure 4), preventing linear regression analyses. However, the main point is that proliferation drops after P7 but both proteins peak later. The authors need to discuss this

discrepancy more clearly.

Author: Thank you for raising this concern. We have adjusted the title of the section to better reflect the data as:

CSF-1 and IL-34 protein and transcript levels increase throughout development

Author: We have also added a section to our discussion to more explicitly contextualize that our arguments for additional regulation, reflecting the discrepancy between high adulthood expression of CSF-1R ligands and minimal microglial proliferation.

Lines 443-446:

Although both CSF-1 and IL-34 are potent microglial mitogens in *in vitro* assays, and while both increase throughout brain development, the fact that they remain high well into adulthood, when microglial proliferation is minimal, implies that they are not solely responsible for developmental microglial proliferation

5, To compare the RNAscope distribution of Il34 and Csf1 (Figure 4) with the expected producing cell, based on their CellChat analysis (Figure 2), the authors show the expression of these two transcripts on the UMAP of the different clusters. However, there seems to be a major difference with the RNAscope data: CellChat identifies IL34 in fibroblasts, pericytes and Cajal Retzius cells, but the RNAscope shows labeling in what seems to be the granule cell layer (Figure 4D). This lack of correlation, if confirmed, could suggest the lack of robustness of the CellChat analysis or a problem with the Joglekar database.

Author: Thank you for noting this. We have addressed this by including new single cell data. Based on our ELISA data and other published work (Devlin et al. 2024), we suspect that P7 may be too early to find such robust expression in neurons and that Il34 expression dramatically increases between P7 and P10. As a result, neuronal Il34 expression may not be evident in the P7 Joglekar dataset. We, therefore, now have included an adult neuron transcriptomics dataset as a supplementary approach, highlighting high Il34 expression in inhibitory and excitatory neuronal populations in adulthood to enhance clarity and ensure that we are not stating that neurons do not express Il34 as has been established in previous work.

Lines 324-340:

The sparse expression of *Il34* in the scRNAseq dataset used for CellChat analyses contrasts with the robust CA1 *Il34* transcript expression at P10 that we observed with RNAscope. Two factors may explain this discrepancy. First, *Il34* may be highly temporally regulated and dramatically increase between P7 and P10, analogous to the increase between P8 and P14 that occurs in the anterior cingulate cortex, nucleus accumbens and amygdala (Devlin et al., 2024). Second, scRNAseq may not optimally capture transcriptional data from neurons. Specifically, the fragility and interconnectedness of neurons may render them susceptible to damage/stress artifacts from the enzymatic and mechanical dissociation techniques used (Joglekar et al., 2021). Great care must be taken in neuronal isolation to avoid such damage/stress (Cuevas-Diaz Duran, Gonzalez-Orozco, Velasco, & Wu, 2022). Thus, we assessed transcriptional profiles of adult (>P53) mouse neurons from a preparation optimized for neuronal viability and survival (Yao et al., 2021). Neurons were isolated from the hippocampus (HIP), parasubiculum, postsubiculum, presubiculum (PAR-POST-PRE), subiculum and prosubiculum (SUB-ProS) (Fig. S5A). We clustered neurons and broadly identified excitatory and inhibitory neurons based on *Slc17a1* and *Gad1* expression, respectively (Fig. 5H, Fig. S5B-C) and found that both express robust levels of *Il34*, with greater levels found in inhibitory populations (Fig. 5I-J) of the adult hippocampus.

6, Another potential issue that should be acknowledged is that the CellChat analysis is based on a single time point (P7), and thus interactions of microglia and other cells over time may be missed.

Author: Thank you for acknowledging this distinction. We have amended the text to reflect the limitations of the current method in our discussion.

Lines 435-441:

Additionally, although we selected a single-cell RNA sequencing dataset at developmental

microglial proliferation for our CellChat analyses, we cannot discount the possibility that transient progenitor cell populations or cellular states that disappear prior to P7, as has been described throughout embryogenesis (La Manno et al., 2021), initiate microglial proliferation, and we simply observed the tail-end of this interaction at P7. Similarly, it may be that cell populations not captured in the single-cell RNA sequencing dataset preparation, due to poor viability or quality control of such cells, could influence microglial proliferation.

7, The authors need to justify the selection of the IL1a and IL1b KO mice as a model to study aberrant microglial proliferation, because: 1, this signaling pathway was not identified in their CellChat analysis; 2, one would expect this pathway to be involved in inflammation-induced proliferation but not basal proliferation; and 3, to be best of my knowledge, the IL1 and IL34 pathways are unrelated. Thus, there is no rationale to assume that IL1-mediated proliferation should be mediated by IL34, nor that IL34 levels would explain proliferation in IL1 KO mice.

Author: Thank you for this comment. We have adjusted the text as follows to provide a clearer rationale for using IL1 knockouts as models of abnormal developmental proliferation and to hopefully better contextualize our interpretations results.

Lines 344-357:

Considering the overlap of microglial proliferation with the increasing expression of *Csf1* and *Il34* throughout development, and the ability of both factors to boost proliferation in culture, we next aimed to explore whether the expression of these mitogens is altered alongside aberrant developmental microglial proliferation. CSF-1R “loss-of-function” models have developmental abnormalities and lack microglia (B. P. Hammond et al., 2021). CSF-1 and IL-34 are critical for microglia survival and, therefore, challenging to disentangle their proliferative and survival-promoting roles in vivo. Fortunately, microglial proliferation has been more widely explored in models of adulthood repopulation following microglial depletion, wherein the depletion of microglia initiates a robust, CNS-wide proliferative phase, analogous to development, that permits microglia to populate the entire CNS (Bruttger et al., 2015; Elmore et al., 2014; Huang et al., 2018). Notably, interleukin-1 receptor-1 signalling partially regulates this proliferation following microglial depletion (Bruttger et al., 2015). Given the similarity of microglial proliferation between repopulation models and development, we reasoned that interleukin-1 signalling may similarly regulate developmental microglial proliferation.

Lines 470-483:

Suitable models for mechanistically exploring aberrant developmental microglial proliferation remain elusive. Here, we used IL-1 α and IL-1 β knockout mice as a model of abnormal proliferation, and while we can adequately assess changes in proliferation, this remains an imperfect model given a lack of interleukin-1 receptor 1 (IL-1R1) expression in microglia (Liu et al., 2019). Thus, any impacts on microglial development are presumably mediated by an additional cell lineage, such as astrocytes, endothelial cells, or neurons. For example, endothelial and epithelial IL-1R1 recruit other immune cells to the CNS in non-developmental contexts, IL-1 α and/or IL-1 β may likewise recruit microglial progenitors to the developing CNS. A lack of either cytokine may delay microglial infiltration and subsequently augment microglial proliferation within the developing CNS. Thus, we cannot discount the possibility that, rather than stemming from altered mitogen levels, the elevated developmental microglial proliferation in the IL-1 α and IL-1 β knockout mice may relate to delayed allometric expansion of microglia within the brain, although we are unable to provide evidence of this in our current study.

7, The statement that “Cx3cl1, rather than directly inducing microglial proliferation, instead recruits microglia to, and/or facilitates transit within, the developing CNS and thus situates microglia within the proliferative niche of the developing CNS” (line 333) is beyond any data provided and excessively speculative. This reviewer is puzzled by the concept of “proliferative niche of the developing CNS” - microglia proliferate throughout the parenchyma, not in proliferative niches. The sentence should be

Author: Thank you for this correction. We agree and the text has been adjusted.

Lines 413-416:

This may, at first glance, appear to conflict with our finding that Cx3cl1 is not mitogenic, however, Cx3cl1, rather than directly inducing microglial proliferation, might instead promote microglia

migration within the developing CNS.

Minor issues:

1. Whether microglia "actively phagocytose synapses" (line 65) has been recently questioned in Eyo and Molofsky Science 2023 (PMID: 37708287) and Pereira-Iglesias et al., Nat Neurosci 2024 (PMID: 39663381), and should be rephrased.

Author: Thank you for bringing these two excellent reviews to our attention. This is an important distinction to keep in mind when studying microglial development. We have reworded this statement to better reflect the current state of the field. Please see changes as follows in lines 63-67.

Line: 67-70

During development, microglia eliminate synaptic material (Paolicelli et al., 2011; Parkhurst et al., 2013; Schafer et al., 2012; Squarzone et al., 2014) and myelin (Djannatian et al., 2023; Hughes and Appel, 2020). However, whether microglia actively sculpt developing neural circuitry remains unclear and has been reviewed elsewhere (Eyo and Molofsky, 2023; Pereira-Iglesias et al., 2025).

2. The sentence on "microglia (...) precipitate depressive-like and anxiety-like behaviors in adulthood (Delpech et al., 2016)" does not reflect the results from said paper, which is largely correlational. The sentence should be rewritten (line 73).

Author: Thank you for this suggestion. We have adjusted the text to reflect better the results of the original paper (lines 69-76)

Line 73-80:

"...although the direct consequences of microglial disruption remain unclear. It may be that the resultant alterations to neural circuitry and activity that stem from microglial disruption have a long-term impact on behaviour. For example, early life stress induced by maternal deprivation in mice throughout the first postnatal weeks yields a transient elevation in microglial densities that normalizes to baseline levels in adulthood. Such stress likewise induces long-term changes in microglial transcriptional state and phagocytic capacity which, given their role in the uptake of synaptic material and myelin throughout development, may contribute to the long-term depressive-like and anxiety-like behaviours of these mice in adulthood (Delpech et al., 2016).

3. There seems to be a grammatical error in the sentence "(...) microglial proliferation in development results from a mitogen directly promotes microglial proliferation" (line 110)

Author: Thank you for noting this. We have corrected this error.

4, There is a typographical error in the sentence "a publicly available scRNAseq datasets" (line 144).

Author: Thank you. This has been corrected.

5, Figure 3: show the Ki67 channel separated, in addition to the merged image.

Author: Thank you for this request. We have altered the representative images in Figure 3C to show each channel separately.

6, The sentence "that CSF-1 promotes equivalent levels of proliferation with or without TGF- β 2, CSF-1 likely promotes proliferation independently from survival." is confusing. What experimental groups do the authors compare to test the effect of CSF "with or without TGF β 2"?

Author: Thank you for this comment. We have removed this line.

7, Figure legends should state the meaning of asterisks and the statistic test used in each comparison.

Author: Thank you for this request. We have updated the figure legends to include this information.

8, The color code of the UMAP clusters is not explained in the figure legend (Figure 2 and 5).

Author: Thank you for this request. We have adjusted figure legends to reflect that each colour in the UMAP represents a distinct cluster of a cell lineage or cellular transcriptional state.

Reviewer 3: SUMMARY OF THE ADVANCE MADE IN THIS PAPER AND ITS POTENTIAL SIGNIFICANCE TO THE FIELD

In the study the authors investigated the factors driving microglia proliferation across key developmental stages. Using immunofluorescence, they characterized microglia density and proliferation in the cortex and hippocampus from P0 to P30. To identify potential drivers of proliferation, the authors analyzed a previously published transcriptomic dataset, pinpointing 12 candidate ligand-receptor pairs. In vitro assays revealed that two ligands appreciably enhanced microglia proliferation (CSF1 and IL-34). These ligands were subsequently evaluated in whole brain homogenates using ELISA and in tissue slices via immunofluorescence and RNAscope to assess spatiotemporal expression in the cortex and hippocampus. These experiments identified IL-34 as the only ligand with consistent and significant expression across all developmental time points in each assay. However, the authors propose that other identified ligands may act synergistically or indirectly contribute to creating a permissive environment for microglia proliferation.

This study adds modestly to prior literature on the regulation of microglia development and proliferation by CSF1R signaling. While seminal work has identified that CSF1 and IL-34 contribute to microglia numbers and survival during development, here the authors question if these ligands also induce proliferation. As the authors' work makes clear, it is challenging to distinguish ligand effects on proliferation when there is also a requirement for survival. Also, there are concerns about the author's interpretation about the source of ligands. Multiple prior studies show that IL-34 is highly expressed by neurons in the developing brain, but that is not evident from the CellChat analysis using the published transcriptomic dataset, raising concerns about the dataset used.

SUGGESTIONS TO AUTHORS

There are some areas in the methods, figures, and experimental design that require clarification. Addressing these concerns will improve the manuscript's clarity and reproducibility.

Major Comments:

1) Microglia Marker Limitations (Figure 1): Iba1 expression is not exclusive to microglia, making it difficult to definitively attribute proliferation or increased density to microglia alone. The observed proliferation might involve myeloid cells recruited from outside the microglia population, raising questions about whether the expansion is solely due to resident microglia or includes recruited macrophages. This issue should be addressed by using a more specific microglia marker such as Sall1 or P2ry12.

Author: This is a very important point. Thank you for bringing it up. We have included an analysis of Tmem119 and Iba1 expression in a new supplemental figure (Fig S1) that demonstrates Tmem119, and potentially other canonical microglial markers, may be insufficient to track microglia specifically at these early postnatal timepoints.

Lines 122-133:

It must be noted that Iba1 is expressed in many myeloid cell populations and not restricted to microglia. Thus, we first assessed whether the canonical microglial marker Tmem119 may suitably track developmental microglial proliferation. However, we found minimal expression of Tmem119 between P0 and P10 in both regions, though by P14, approximately 70% of all Iba1 cells in the somatosensory cortex, and approximately 50% of all Iba1 cells in CA1, expressed Tmem119 (Fig. S1A-C). A similar finding has been reported at the transcript level, wherein Tmem119 and other canonical microglial markers were not found at P4/P5 in the developing brain (Hammond et al., 2019). Together, these findings suggest that Tmem119, and potentially other canonical microglial markers, may not be ideal for tracking microglia in early postnatal development. Thus, given the

robust Iba1 expression and parenchymal location of these Iba1- labelled cells, we will refer to them as microglia throughout this work.

2) Significance of Cell Proliferation (Figure 3): The authors identify CSF1, IL-34, and ANGPTL4 as ligands that promote proliferation in vitro. However, CellChat predicts minimal "communication probability" for CSF1 and ANGPTL4. This discrepancy between in vitro results and computational predictions should be addressed in the discussion, particularly in light of the high communication probability of Ptn, but limited impact on proliferation thus no further study. As noted above, the cell type enrichment from this analysis does not align with prior published work showing high enrichment of IL-34 in neurons.

Author: Thank you for noting this. We have elaborated on this in the discussion to enhance clarity and interpretation of our CellChat results.

Lines 403-407

However, CellChat predicts any potential ligand-receptor interactions, not solely those that may impact proliferation. It is entirely possible that the predicted non-mitogenic interactions regulate other developmental microglial dynamics such as recruitment or migration within the developing CNS. Such non-mitogenic functions may indirectly facilitate developmental microglial proliferation.

Author: Thank you for highlighting the lack of Il34 expression in neuron populations in our CellChat analysis. We agree and have now added clarification and data to address this point. We have also included an analysis on a publically available adulthood dataset which demonstrates robust Il34 expression in neurons

Lines 324-340:

The sparse expression of *Il34* in the scRNAseq dataset used for CellChat analyses contrasts with the robust CA1 *Il34* transcript expression at P10 that we observed with RNAscope. Two factors may explain this discrepancy. First, *Il34* may be a highly regulated temporally, dramatically increasing between P7 and P10, similar to an analogous increase between P8 and P14 in the anterior cingulate cortex, nucleus accumbens and amygdala (Devlin et al., 2024). Second, scRNAseq may not optimally capture transcriptional data from neurons. Specifically, the fragility and interconnectedness of neurons may render them susceptible to damage/stress artifacts from the enzymatic and mechanical dissociation techniques used (Joglekar et al., 2021). Great care must be taken in neuronal isolation to avoid such damage/stress (Cuevas-Diaz Duran et al., 2022). Thus, we assessed transcriptional profiles of adult (>P53) mouse neurons from a preparation optimized for neuronal viability and survival (Yao et al., 2021). Neurons were isolated from the hippocampus (HIP), parasubiculum, postsubiculum, presubiculum (PAR-POST-PRE), subiculum and prosubiculum (SUB-ProS) (Fig. S5A). We clustered neurons and broadly identified excitatory and inhibitory neurons based on *Slc17a7* and *Gad1* expression, respectively (Fig. 5H, Fig. S5B-C) and found that both express robust levels of *Il34*, with greater levels found in inhibitory populations (Fig. 5I-J) of the adult hippocampus.

3) Methodology and Media Use (Figure 3): Lines 182-186 could be rewritten as the methodology for switching out different medias and timing is confusing. In addition, the use of MGM media for only a subset of the mitogens identified raises some concerns:

* Since TGFB2 and CSF1 are already included in the serum-free media, this potentially confounds the interpretation of additional mitogen effects. Ie---the impact of other mitogens added may not be appreciated in the presence of CSF1 in the serum-free media, since it was shown to be sufficient to increase microglia proliferation in figure 3C. Why not compare all mitogens across the same media conditions? If TGFB2 and CSF1 are deemed essential for microglial survival, how do microglia survive in the base media, which lacks these factors?

Author: This is an important point. We have redone these experiments as suggested, which are now found in Fig 3B and tested in media lacking CSF-1 and TGF-B2. We have also revised our explanation for this to enhance clarity.

Lines 225-239:

Given that two of these factors—CSF-1 and TGF-B2—are essential survival factors for microglia we first conducted a viability assessment with calceinAM and propidium iodide on microglia with an

identical timeline of five days of culture in microglial growth media (GM; contains 10 ng/mL CSF-1, 2 ng/mL TGF- β 2), followed by three days of growth in base media alone (no CSF-1 or TGF- β 2), or base media with TGF- β 2, CSF-1, or TGF- β 2 and CSF-1 (GM). Both TGF- β 2 or CSF1 promoted similar levels of microglial viability to microglial growth media containing both CSF1 and TGF- β 2 (Fig. S3E). However, given that we wished to assess microglial proliferation of each factor individually, we elected to test all predicted signalling factors in microglial base media (no CSF-1 or TGF- β 2). We first cultured microglia for five days in microglial growth media (with CSF-1 and TGF- β 2) before performing treatments of each factor at 10 ng/mL in microglial base media for an additional 72 hours. Although a lack of both growth factors reduces microglial viability (Fig. S3E), we deemed their viability throughout the treatment period sufficient for these analyses. Of the predicted signalling factors, only CSF-1, and IL-34 to a lesser extent, boosted microglial proliferation (Fig. 3B).

* Would it be possible to clarify the significance of comparing MGM in the 3C figure? Was this media used in any of the assays?

Author: Thank you for mentioning this. We have removed this portion of Figure 3 as it was now unnecessary in lieu of other changes we have made.

* Please clarify the concentrations of TGF β 2 and CSF1 in the serum-free condition versus the additional amounts included in the experimental conditions. While Figure 3D specifies the concentration of the added mitogen, this information is absent for Figure 3C, making comparisons difficult.

Author: Thank you for this suggestion. We now clarify TGF β 2 and CSF1 concentrations in the text and figure.

* The axes in Figures 3C and 3D are not consistent. Standardizing the axes would facilitate easier comparison between the data.

Author: We have removed both of these figures in lieu of other changes to Figure 3.

* Why is CD11b used as the marker for identifying "microglia" in this figure, while IBA1 is used in other figures? CD11b is less specific to microglia compared to IBA1 and may also label other myeloid cells.

Thank you for noting this important distinction. Cd11b provided a better signal to noise ratio for cultured microglia. We have included a Tmem119 colabeling experiment in Fig 3A, C to demonstrate that nearly 100% of these Cd11b⁺ cells express a canonical microglial marker.

Line 212-217:

We isolated primary mouse microglia from the P5-P7 cortex by immunopanning (Fig. 3A), which yielded highly pure (on average 95.4%) CD11b⁺ microglial cultures, of which nearly 100% were also Tmem119⁺, by day 7-8 *in vitro* (Fig. S3A-C). Considering these cells were isolated from brains with low microglial Tmem119 expression (Fig. S1A-B), the inclusion of TGF- β 2 in the serum-free media may promote maturation of microglia *in vitro* as it does *in vivo* (Utz et al., 2020).

* In Figure 3D, CSF1, MGM, and IL34 are not listed, despite their relevance and further exploration of impact on microglial proliferation. Could the authors clarify why these ligands/MGM were omitted from this graph? Additionally, there is no rationale provided for the specific selection of ligands for the dose-response curves in Figures 3E and 3F. Including a brief explanation for the choice of ligands would be helpful. Ie—why was ANGPTL4 not examined even though it demonstrates the highest fold change out of all the ligands?

Author: Thank you for this comment. We have redone some of these analyses as described above and did not find ANGPTL4 to be a microglial mitogen (Fig S4D). The results are updated in Fig. 3B. The selection of CSF-1 and IL-34 for the dose response curves was predicated on the fact that they were the only two factors that were mitogenic in Fig. 3B.

4) Confounding Results (Figure 3): The claim that IL-34 increases proliferation should be

reconsidered, as the data for this is not clearly presented (IL-34 is not depicted in figure 3C or 3D compared to other ligands). The overlap of CSF1 present in the serum-free media could confound results of the addition of IL34 since they both act on the same receptor. The authors should discuss how this may influence the interpretation of the role of IL34 on proliferation.

Author: Thank you for this important comment. We have rephrased this section as it was misleading before. In the IL-34 dose response of Figure 3D-E, CSF-1 was excluded from our base media. Therefore, IL-34 was the only CSF-1R ligand present in the IL-34 dose response, and CSF-1 was the only CSF-1R ligand present in the CSF-1 dose response. We believe our conclusion that IL-34 increases proliferation is now accurately reflected by the data presented in Fig. 3B-D. We have adjusted the text as follows to enhance clarity in methodology and results.

Lines 242-245:

Both CSF-1 and IL-34 share a receptor—CSF-1R—and so the differences in impacts on proliferation may suggest differential impacts of each at different concentrations. To test this, we conducted dose response experiments for each CSF-1 and IL-34 (Fig 3C-D) in a microglial base media that contained only TGF- β 2 to enhance survival (Fig S3C).

5) Inflammation Model (Figure 6): The rationale for using IL-1B and IL-1a KO mice to model aberrant inflammation seems insufficient, especially since the authors cite literature that microglia may not express the receptors for these cytokines, suggesting indirect effects. There is a potential inconsistency between the interpretation of proliferation and density in the IL-1B KO model. Increased proliferation with decreased density contradicts earlier claims from Figure 1 that attributed increased density to proliferation. Further clarification is needed regarding how these findings align.

Author: Thank you for this comment. We have adjusted the text as follows to provide a clearer rationale for using IL1 knockouts as models of abnormal developmental proliferation and to hopefully better contextualize our interpretations results.

Lines 344-357:

Considering the overlap of microglial proliferation with the increasing expression of *Csf1* and *Il34* throughout development, and the ability of both factors to boost proliferation in culture, we next aimed to explore whether the expression of these mitogens is altered alongside aberrant developmental microglial proliferation. CSF-1R “loss-of-function” models have developmental abnormalities and lack microglia (B. P. Hammond et al., 2021). CSF-1 and IL-34 are critical for microglia survival and, therefore, challenging to disentangle their proliferative and survival-promoting roles in vivo. Fortunately, microglial proliferation has been more widely explored in models of adulthood repopulation following microglial depletion, wherein the depletion of microglia initiates a robust, CNS-wide proliferative phase, analogous to development, that permits microglia to repopulate the entire CNS (Bruttger et al., 2015; Elmore et al., 2014; Huang et al., 2018). Notably, interleukin-1 receptor-1 signalling partially regulates this proliferation following microglial depletion (Bruttger et al., 2015). Given the similarity of microglial proliferation between repopulation models and development, we reasoned that interleukin-1 signalling may similarly regulate developmental microglial proliferation.

Lines 470-483:

Suitable models for mechanistically exploring aberrant developmental microglial proliferation remain elusive. Here, we used IL-1 α and IL-1B knockout mice as a model of abnormal proliferation, and while we can adequately assess changes in proliferation, this remains an imperfect model given a lack of interleukin-1 receptor 1 (IL-1R1) expression in microglia (Liu et al., 2019). Thus, any impacts on microglial development are presumably mediated by an additional cell lineage, such as astrocytes, endothelial cells, or neurons. For example, endothelial and epithelial IL-1R1 recruit other immune cells to the CNS in non-developmental contexts, IL-1 α and/or IL-1B may likewise recruit microglial progenitors to the developing CNS. A lack of either cytokine may delay microglial infiltration and subsequently augment microglial proliferation within the developing CNS. Thus, we cannot discount the possibility that, rather than stemming from altered mitogen levels, the elevated developmental microglial proliferation in the IL-1 α and IL-1B knockout mice may relate to delayed allometric expansion of microglia within the brain, although we are unable to provide evidence of this in our current study.

6) Strain Discrepancies: The authors use different mouse strains for ELISA assays at different time points (P0, P7, P30). It would be helpful to clarify why this approach was chosen and how strain-specific differences might influence results. Furthermore, the use of the iDTR mouse line is mentioned in methods, but its specific role and representation in the data/main text are not clearly outlined. Why are both B6 and iDTR mice used and at different timepoints?

Author: Thank you for noting this. We have removed all iDTR mice these analyses and replaced with C57BL/6 mice.

7) In Vitro vs In Vivo Mouse Strain Differences: The authors use cells from CD1 mice for in vitro experiments, while in vivo experiments utilize B6/iDTR mice. It would be valuable to explain why these strains are used differently and whether strain variability could impact results and interpretation across in silico, in vitro, and in vivo assays.

Author: Thank you for this comment. We have adjusted the text to read as follows to distinguish between C57 and CD1. We have redone all iDTR analyses with C57 mice.

Lines 201-208:

To explore whether the ligand-receptor pairings predicted to signal onto P7 microglia may boost microglial proliferation, we elected to test these factors on serum-free, primary microglia isolated from CD1 mice. Although primary CD1 microglia display subtly elevated inflammatory responses in response to LPS stimulation relative to C57BL/6 counterparts, both *in vitro* and *in vivo* (Nikodemova & Watters, 2011), the greater litter sizes, and thus greater yield of microglia, made them a suitable model for our cell culture experiments. Further, the use of outbred mouse stocks for studying immunological mechanisms has been suggested to be a critical “next step” in translating findings from the bench to bedside (Enriquez et al., 2020).

Minor Comments:

8) Figure 1 (Y-Axis Labeling): The Y-axis labels for panels C and F are unclear. Clarifying the measurement scale and confirming whether 0.2 microglia per mm² is correct would help prevent misinterpretation.

Author: Thank you for noting this. We have adjusted this accordingly and confirmed that our initial number was incorrect as we had multiplied by the incorrect correction factor.

9) Significance Markers: It may be helpful to report exact p-values rather than using stars for significance in the graphs to improve clarity and rigor.

Author: Thank you for this suggestion. We have included a line in each of the figure legends to specify the a limit for significance as follows:

p<0.05, **p<0.01, *p<0.001, ****p<0.0001.*

10) Methodological Clarification (Figure 1): The figure legend should specify the number of sections per brain examined, which is currently missing in the methods section.

Author: We have added this to the figure legend as follows:

Counts of microglia and proliferating microglia were averaged from 3-4 images in either the somatosensory cortex or CA1.

11) Isolated Cell Identity (Figure 3): How are isolated microglia being confirmed? Is CD11b the sole marker used? CD11b is not microglia-specific and can be expressed by other myeloid cells, such as macrophages. Incorporating additional markers like P2RY12 or Sall1 could strengthen the identification of microglia. Additionally, the morphology of the cells shown in the image does not align with the characteristic ramified structure typically observed in microglia under homeostatic conditions.

Author: Thank you for noting this. The morphology of primary cells dramatically changes when removed from the structural and molecular milieu of the brain, and so, cells like microglia often fail to completely recapitulate the ramified morphology of homeostatic microglia in the brain. However, we have used Tmem119 colabeling with CD11b to confirm microglial cell identity in Fig. S2.

12) In Situ vs Protein (Figure 5): There seems to be a discrepancy between in situ staining and protein quantification, for the time points used. More clarity on this would be beneficial.

Author: Thank you for noting this. We have included new data to ensure we are comparing similar timepoints (please see Fig. 1, Fig 4, Fig 6).

13) IL-34 Staining (Figure 5): It would be useful to define the threshold for positive IL-34 staining and provide a more detailed explanation of the staining pattern in the figure legend. It is listed in the methods, but it would be helpful to have in the results or figure legend description as well.

Author: Thank you for this request. We have added this to our results and methods sections.

Lines 275-276:

We counted cells as positive if they contained five or more *Csf1* or *Il34* transcripts.

Lines 748-749:

Thresholds for *Csf1* or *Il34* positive nuclei were determined based on the fluorescent intensity corresponding to 5 transcripts within nuclei.

14) Image Analysis (Figure 5): The methods should specify how many images per mouse were analyzed/averaged and how regions of interest in the cortex and hippocampus were selected. Providing these details would improve reproducibility.

Author: Thank you for this request. We have adjusted this in the methods as follows:

Lines 749-756

For the developmental timeline of microglial proliferation, we counted and averaged a total of four images—two from each hemisphere—from within CA1 and layers II-IV of the somatosensory cortex. For RNAscope analyses, we averaged counts in ROIs of CA1 and layers II-IV of the somatosensory cortex from each hemisphere. For the timeline of developmental microglial proliferation in IL-1 knockouts, we averaged counts of microglia and proliferative microglia throughout all regions of the hippocampus and layers II-IV of the somatosensory cortex across ROIs in 1-2 brain sections.

Second decision letter

MS ID#: dev.204610R1

MS Title: CSF-1R ligands promote microglial proliferation but are not the sole regulators of developmental microglial proliferation

Authors: Brady P. Hammond; Sameera Zia; Eugene Hahn; Margarita Kapustina; Tristan Lange; Sarah Friesen; Rupali Manek; Kelly V. Lee; Adrian Castellanos-Molina; Floriane Bretheau; Mark S Cembrowski; Bradley J. Kerr; Steve Lacroix; Jason Plemel

Article Type: Research Article

Dear Dr Plemel,

I am delighted to tell you that your manuscript has been accepted for publication in Development, pending our standard publication integrity checks.

Reviewer 1

The authors have adequately addressed the issues that were raised.

Reviewer 2

All my concerns have been addressed

Reviewer 3

The authors have carefully addressed the prior comments and provided new data to support their conclusions. I have no further concerns.